# A type II protein arginine methyltransferase regulates merozoite invasion in *Plasmodium falciparum*

Amuza Byaruhanga Lucky [1,5], Chengqi Wang [2,5], Min Liu[1,3,5], Xiaoying Liang[1], Hui Min[1], Qi Fan[4], Faiza Amber Siddiqui[1], Swamy Rakesh Adapa[2], Xiaolian Li[1], Rays H. Y. Jiang[2], Xiaoguang Chen [3], Liwang Cui [1] & Jun Miao [1✉]

Protein arginine methyltransferases (PRMTs) regulate many important cellular processes, such as transcription and RNA processing in model organisms but their functions in human malaria parasites are not elucidated. Here, we characterize PfPRMT5 in *Plasmodium falciparum*, which catalyzes symmetric dimethylation of histone H3 at R2 (H3R2me2s) and R8, and histone H4 at R3 in vitro. *PfPRMT5* disruption results in asexual stage growth defects primarily due to lower invasion efficiency of the merozoites. Transcriptomic analysis reveals down-regulation of many transcripts related to invasion upon *PfPRMT5* disruption, in agreement with H3R2me2s being an active chromatin mark. Genome-wide chromatin pro-filing detects extensive H3R2me2s marking of genes of different cellular processes, including invasion-related genes in wildtype parasites and *PfPRMT5* disruption leads to the depletion of H3R2me2s. Interactome studies identify the association of PfPRMT5 with invasion-related transcriptional regulators such as AP2-I, BDP1, and GCN5. Furthermore, PfPRMT5 is asso-ciated with the RNA splicing machinery, and *PfPRMT5* disruption caused substantial anomalies in RNA splicing events, including those for invasion-related genes. In summary, PfPRMT5 is critical for regulating parasite invasion and RNA splicing in this early-branching eukaryote.

[1] Department of Internal Medicine, Morsani College of Medicine, University of South Florida, Tampa, FL 33612, USA. [2] Center for Global Health and Infectious Diseases, Department of Global Health, University of South Florida, Tampa, FL 33612, USA. [3] Department of Pathogen Biology, School of Public Health, Southern Medical University, Guangzhou, Guangdong 510515, China. [4] Dalian Institute of Biotechnology, Dalian, Liaoning, China. [5] These authors contributed equally: Amuza Byaruhanga Lucky, Chengqi Wang, Min Liu. ✉email: jmiao1@usf.edu

*P*lasmodium falciparum is a unicellular parasite that causes the most virulent human malaria, responsible for about half a million deaths annually[1]. In the human host, the parasite's cyclic multiplication and destruction of the red blood cells (RBCs) are responsible for the clinical manifestations and morbidity of malaria. The intraerythrocytic development cycle (IDC), beginning from the merozoite's invasion of an RBC and progressing through the morphologically distinct ring, trophozoite, and schizont stages, completes in about 48 h. The IDC is orchestrated by a tightly regulated transcription program dictating waves of stage-specific gene expression[2,3]. The transcription program is multi-layered, including specific transcription factors such as the AP2-domain proteins and epigenetic mechanisms[4,5]. Epigenetic regulation is accomplished through remodeling the chromatin structure between the active and silent state by mechanisms such as nucleosome positioning, incorporation of variant histones, and reversible modifications of histones. *P. falciparum* histones carry a myriad of post-translational modifications (PTMs)[6–11], consistent with the large compendium of histone modification enzymes and histone PTM-binding proteins encoded by the parasite genome[4]. As an example of multifactorial coordination, genes involved in merozoite invasion are regulated by specific AP2 domain-containing transcription factors AP2-I[12], PfGCN5[13], and PfBDP1[14]. Genome-wide profiling of PTMs and functional studies of proteins mediating the deposition, removal, and binding of PTMs provide an ever-improving understanding of epigenetic mechanisms in numerous aspects of *Plasmodium* biology, laying the necessary foundation for developing "epidrugs" for malaria chemoprevention.

Protein PTMs, such as acetylation, methylation, phosphorylation, and ubiquitylation, are central to epigenetics[15,16]. In recent years, arginine methylation has attracted increasing attention as a key PTM, mainly due to the discovery of its link to human diseases such as cancer and the recognition of its value in therapeutic development[17]. A family of protein arginine methyltransferases (PRMTs) catalyzes the deposition of active or repressive histone marks, regulating gene expression[18,19]. In addition, they also deposit arginine methylation on non-histone substrates, regulating essential cellular processes such as transcription, cell signaling, mRNA translation, and pre-mRNA splicing[18,20]. Because of the availability of two-terminal nitrogens for methylation, arginine can be methylated once (monomethylarginine – MMA) and twice (asymmetric $\omega$-$N^G$, $N^G$-dimethylarginine – aDMA and symmetric $\omega$-$N^G$, $N^{G'}$-dimethylarginine – sDMA). Depending on the types of modification they catalyze, PRMTs are classified into three subgroups: Type I (catalyzing the formation of MMA and aDMA), Type II (catalyzing the formation of MMA and sDMA), and Type III (catalyzing the formation of MMA only)[18,21]. The human genome encodes nine PRMTs, of which PRMT5 is the predominant Type II methyltransferase. Intriguingly, PRMT5 can be a corepressor and a coactivator, depending on the Arg residues it methylates. PRMT5 deposits the symmetric dimethylation marks on histone H3R8 and the R3 motifs present at H2A and H4, which are associated with transcriptional repression[18,22]. Conversely, PRMT5 also symmetrically methylates H3R2, which recruits coactivator complexes and is highly correlated with the active mark H3K4me3 at active promoters[23,24]. Moreover, PRMT5 interacts with various partners to specifically methylate non-histone substrates, such as the components in the RNA splicing pathway. Thus, PRMT5 deletion has pleiotropic effects, resulting in defects in RNA splicing, cell differentiation, and development[25–27].

With the recognition of PRMTs as important therapeutic targets for human diseases[20,28], the significance of PRMTs in the biology of protozoan parasites has received increasing attention[29,30]. The enzymatic activities, substrates, and functions of five *Trypanosoma brucei* PRMTs, including the Type I PRMTs (TbPRMT1 and 6), Type II PRMT5, and Type III PRMT7, have been investigated, and the knockdown of TbPRMT1 and TbPRMT5 by RNAi resulted in growth defects[29,31–33]. TbPRMT5 was identified as a divergent type II PRMT5 since it exclusively localizes in the cytoplasm and was associated with several proteins involved in a methylation-related function[34]. *Toxoplasma gondii* PRMT1 and CARM1 catalyzed histone H4R3 and H3R17 methylation in vitro and attributed to cell cycle regulation and gene activation, respectively[35,36]. Three putative PRMTs (PfPRMT1, PRMT4/CARM1, and PfPRMT5) have been identified in *P. falciparum*[29,37,38], but their functions remain uncharacterized. PfPRMT1 is a Type I PRMT located in the nucleus and cytoplasm, with activities toward histones and non-histone substrates[37]. PfPRMT5 was found to be associated with the spliceosome core protein PfSmD1[39], suggesting that PfPRMT5 may regulate splicing in malaria parasites.

In this study, we characterized the Type II PRMT, PfPRMT5, in *P. falciparum*. We provide solid evidence to establish PfPRMT5 as a critical regulator of invasion. We also demonstrate that the PfPRMT5 mediates symmetric dimethylation of histone H3R2 (H3R2me2s), and its genome-wide distribution is consistent with H3R2me2s serving as an active chromatin mark required for efficient transcription of invasion-related genes. In addition, PfPRMT5 disruption caused considerable changes in RNA splicing in the parasite, suggesting the role of PfPRMT5 in regulating splicing.

## Results

**PfPRMT5 is a putative type II PRMT.** Of the three PRMTs (PfPRMT1, PfPRMT4, and PfPRMT5) identified in the *P. falciparum* genome[37], *PfPRMT5* (PF3D7_1361000) is an intron-less gene of 2175 bp, encoding a protein of 724 amino acids (aa) with a predicted molecular mass of 85.6 kDa. The conserved catalytic core domain, located in the C-terminus (447–668 aa), bears all signature motifs for SAM-dependent methyltransferases, including the SAM-binding domain (motif I, Post I, and III), a double-E loop critical for substrate recognition and methylation, and the THW loop (Supplementary Fig. 1a). The long N-terminal domain is highly conserved in all *Plasmodium* PRMT5s but has no similarity to known protein domains. The PfPRMT5 catalytic domain displays 44/64%, 37/56%, and 25/39% identity/similarity with the PRMT5 homologs from *H. sapiens* (HsPRMT5/JBP1), *S. pombe* (Skb1), and *T. brucei* (TbPRMT5), respectively (Supplementary Fig. 1b). The PfPRMT5 F390 residue, conserved in all PRMT5 homologs, specifies the type II enzyme activity since the mutation of this residue to Met in the *C. elegans* PRMT5 changed it to a Type I PRMT with aDMA activity[40]. Prediction of PfPRMT5 structure by aligning it with the crystal structure of HsPRMT5 resulted in a confidence C-score of −0.98 and a TM-score of 0.831, indicating high-quality prediction and high structural similarity between PfPRMT5 and HsPRMT5 (Supplementary Fig. 2).

**PfPRMT5 is constitutively expressed and localized in both cytoplasm and nucleus.** The *PfPRMT5* transcript has been identified in transcriptomic studies, which showed a peak level in the early stages of the IDC[2,41–44]. Real-time RT-PCR result was compatible with the findings from these studies, showing peak *PfPRMT5* mRNA levels in the ring and early trophozoite stages (Supplementary Fig. 3a). The transcription start site (TSS) and the polyadenylation site of the *PfPRMT5* mRNA were determined by RLM-RACE and 3'-RACE using RNA from asexual bloodstage parasites (Supplementary Fig. 3b). All 12 clones sequenced for the 5'-RLM-RACE mapped the 5' end to −132 bp upstream of

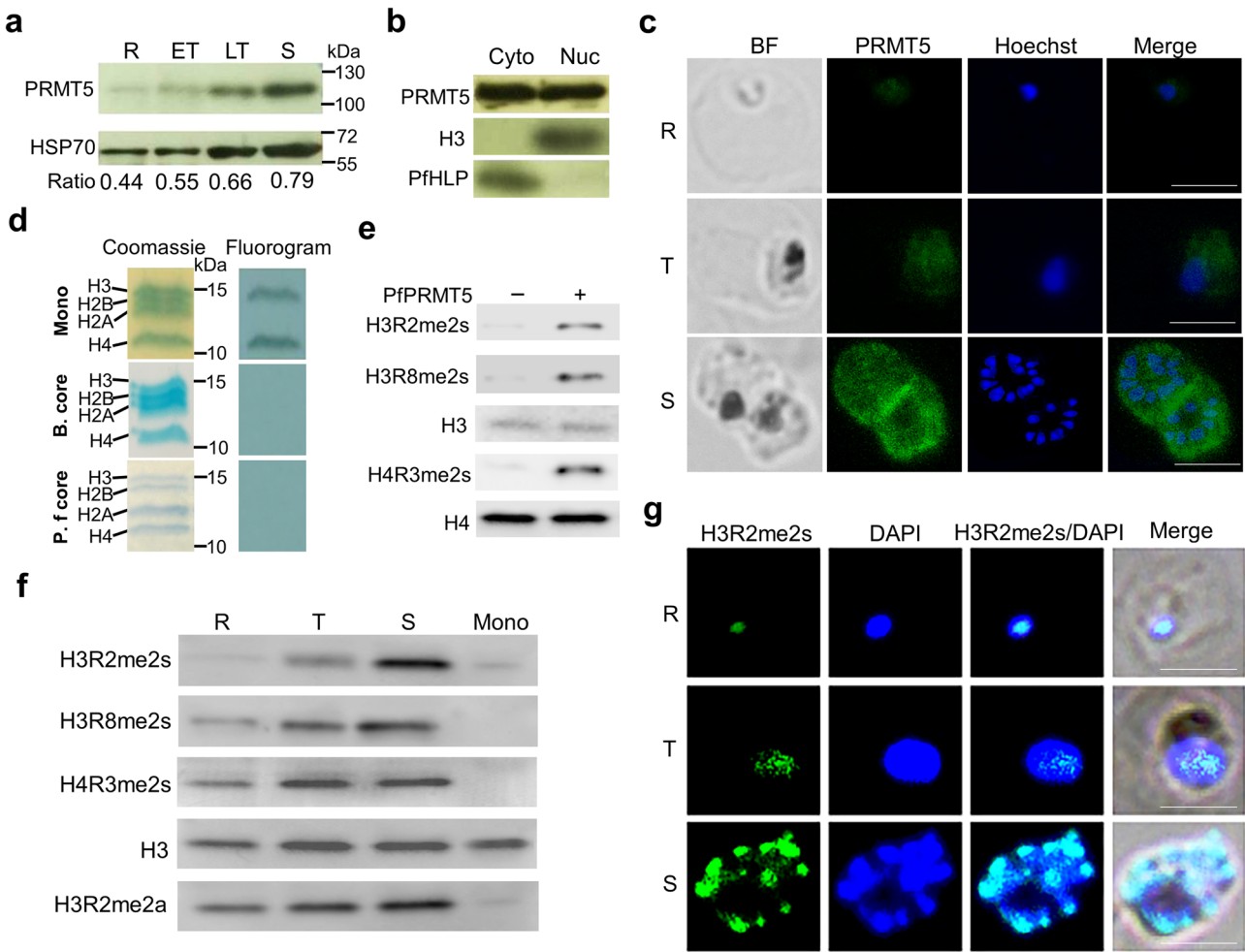

**Fig. 1 PfPRMT5 expression, localization, and its histone modifications during IDC. a** PfPRMT5 expression at different stages of the IDC was analyzed by Western blots with anti-Protein C antibodies (targeting PTP tag) to detect the full-length PfPRMT5-PTP (~ 106 kDa) in the PfPRMT5::PTP parasite line. PfHSP70 was used as a loading control. R: ring, ET: early trophozoite, LT: late trophozoite, and S: schizont. The relative expression level of PfPRMT5 was normalized with PfHSP70. **b** Identification of PfPRMT5 in the parasite cytoplasmic (Cyto) and nuclear (Nuc) fractions (upper panel). Anti-H3 antibodies (middle panel) and anti-PfHLP antiserum (lower panel) are used as nuclear and cytoplasmic markers, respectively. **c** IFA with anti-Protein C and FITC-conjugated anti-rabbit IgG as primary and secondary antibodies, respectively, to detect the localization of PfPRMT5 in the PfPRMT5::PTP parasite line. Nuclei were stained with Hoechst 33342. The size of the scale bar is 5 μm. **d** Endogenous PfPRMT5 was used in the methylation assay with human mononucleosomes, bovine, and *P. falciparum* core histones as the substrates, and the reactions were separated by SDS-PAGE (15% gel). Left panel: Coomassie blue-stained gel. Right panel: fluorograph. **e** Western blots with anti-H3R2me2s, H3R8me2s, and H4R4me2s antibodies were conducted to detect PfPRMT5-dependent methylation on H3 and H4 in human mononucleosomes. H3 and H4 were used as loading controls. **f** The levels of H3R2me2s, H3R2me2a, H3R8me2s, and H4R4me2s in the WT parasite during the IDC were analyzed by Western blots with respective antibodies. H3 was used as a loading control and human mononucleosome (Mono) was used as a negative control without any modifications in the histones. **g** IFA with anti-H3R2me2s and Alexa fluor 488-conjugated anti-rabbit IgG as primary and secondary antibodies, respectively, to detect the localization of H3R2me2s in the WT 3D7 parasite. Nuclei were stained with DAPI. The size of the scale bar is 5 μm.

the putative ATG codon, consistent with the TSS block of *PfPMRT5* (−120 to −130 bp) identified by RNA-seq[45]. All four clones from the 3'-RACE analysis detected a single poly-adenylation site at 183 bp downstream of the stop codon, the same as determined by amplification-free RNA-seq[46]. These analyses predicted the *PfPRMT5* mRNA to be ~2.5 kb (Supplementary Fig. 3b).

To study PfPRMT5 protein expression, we tagged the C-terminus of the endogenous PfPRMT5 with the PTP tag (Supplementary Fig. 3c). Correct integration of the transfected plasmid at the *PfPRMT5* locus was verified by integration-specific PCR (Supplementary Fig. 3d). Western blot using parasite lysates from PfPRMT5::PTP parasite lines with antibodies against protein C detected a band of ~110 kDa, consistent with the size of the PfPRMT5-PTP fusion protein (Supplementary Fig. 3e).

The tagging of PfPRMT5 with PTP did not cause noticeable changes in parasite growth. PfPRMT5 protein was detected throughout the IDC, but the protein levels increased continuously from the ring to the schizont stage (Fig. 1a). This discrepancy between PfPRMT5 protein and mRNA levels indicates post-transcriptional regulation of *PfPRMT5*.

We used cell fractionation and immunofluorescence assay (IFA) to determine the subcellular localization of PfPRMT5. Analysis of PfPRMT5 by cell fractionation followed by Western blots with antibodies for the nuclear (H3) and cytoplasmic (PfHLP) compartments detected the presence of PfPRMT5 in both fractions (Fig. 1b). In addition, the IFA of PfPRMT5::PTP parasites using anti-protein C antibodies detected PfPRMT5 localization in both the nucleus and cytoplasm and the signals of PfPRMT5 increased during the parasite development (Fig. 1c).

**PfPRMT5 methylates histones H3 and H4 in mononucleosomes**. Histones are among the major substrates for PRMT5[22]. To determine the enzymatic activity of PfPRMT5 on histones, we purified the endogenous PTP-tagged PfPRMT5 using a tandem affinity purification (TAP) procedure and performed in vitro methylation assays. Although the purified PfPRMT5-PTP did not methylate bovine core histones or recombinant *P. falciparum* core histones, it displayed methylase activity towards histones in the human mononucleosomes (Fig. 1d), reminiscent of human PRMT5 which requires its partners (such as MEP50, pICln, kinase RioK1, and Grg4) for its activity and substrate specificity[18,19,47,48], suggesting the unknown partners of PfPRMT5 from the TAP eluate potentially lead to the substrate specificity of PfPRMT5. Like the hsPRMT5[49], PfPRMT5-PTP methylated both histones H3 and H4. The human PRMT5 symmetrically methylates H3R8, H4R3, and H3R2[23,49]. To determine the arginines in histone H3 and H4 modified by PfPRMT5, we performed in vitro methylation reaction with purified PfPRMT5-PTP and human mononucleosomes. Using specific antibodies against H3R2me2s, H3R8me2s, and H4R3me2s in Western blots, we found that PfPRMT5 conferred symmetric dimethylation on all three Arg residues in histones H3 and H4 in vitro (Fig. 1e).

**H3R2me2s is substantially enriched in schizonts**. To investigate the levels of the above three methylarginine marks generated by PfPRMT5 in the parasite during the IDC, we purified histones from the 3D7 parasite at the ring, trophozoite, and schizont stages and performed Western blots with specific antibodies for H3R2me2s, H3R8me2s, and H4R3me2s. Since proteomic analyses of *P. falciparum* histones have identified arginine methylation on H3R8 and H4R3, but not on H3R2[6–11], we wanted to determine whether H3R2me2s is present in *P. falciparum*. By dot blot analysis using synthetic H3 peptide with or without H3R2me2s modification, we confirmed that anti-H3R2me2s but not H3R2me2a antibodies were specific (Supplementary Fig. 4). Western analysis revealed that H3R2me2s was most predominant in schizonts, whereas it was expressed at trace and low levels in the ring and trophozoite stages, respectively (Fig. 1f). In contrast, the abundance of the H3R2me2a mark on the same arginine residue was consistent during the IDC. The two repressive histone marks, H3R8me2s and H4R3me2s, were also detected throughout the IDC. H4R3me2s was similarly abundant in all the stages, whereas the H3R8me2s level gradually increased from the ring to the schizont stage (Fig. 1f). We further evaluated the subcellular localization of these histone marks by IFA (Fig. 1g, Supplementary Fig. 5). At the ring stage, the H3R2me2s signal appeared as a single punctum in the nucleus defined by DAPI staining, while it subsequently expanded to a much larger domain, overlapping with part of the DAPI signal (Fig. 1g). The H3R2me2s abundance reached the highest in schizonts across the asexual life cycle, with the signals appearing as separate puncta, overlapping partially with the individual merozoite nuclei (Fig. 1g). This distribution pattern suggests that the H3R2me2s mark is restricted to certain chromatin regions in the asexual parasites. In contrast, the localization patterns of H3R8me2s and H4R3me2s were distinct, suggesting these active and repressive marks might localize to different nuclear compartments[50,51] (Supplementary Fig. 5).

**Disruption of *PfPRMT5* causes growth and invasion defects**. To elucidate the function of PfPRMT5 in development, we disrupted the *PfPRMT5* gene using a single crossover homologous recombination strategy, disrupting the *PfPRMT5* coding sequence at amino acid 397, deleting 327 aa in the PfPRMT5 C-terminus (Supplementary Fig. 6a). After transfection, drug selection, and parasite cloning, transgenic parasite clones were verified for the

correct integration of the plasmid pHD22Y/ΔPfPRMT5 at the *PfPRMT5* locus by Southern blot analysis (Supplementary Fig. 6b). Two *PfPRMT5*-disrupted lines (ΔPfPRMT5-1 and ΔPfPRMT5-2) resulting from two transfection experiments were used to determine the effects of *PfPRMT5* disruption on asexual parasite development. In vitro growths of the wild-type (WT) and ΔPfPRMT5 lines were compared over three asexual development cycles. Starting at a 0.1% parasitemia, both ΔPfPRMT5 lines grew significantly more slowly than the WT 3D7 (Fig. 2a, one-way ANOVA, $P < 0.01$, $n = 3$). On day 7, the two ΔPfPRMT5 lines only reached ~3% parasitemia compared to ~11% in WT 3D7 parasites. To determine whether the slower parasite proliferation resulted from an altered developmental pattern during the IDC, we monitored the cell cycle progression of highly synchronized parasites every 2 h by differentiating parasite stages in Giemsa-stained blood smears (Fig. 2b). However, the ΔPfPRMT5 lines did not show any noticeable change in the progression of the IDC; the dynamics of the ring, trophozoite, and schizont stages were indistinguishable between the 3D7 and ΔPfPRMT5 parasites. This prompted us to investigate whether the reduced proliferation rates in the ΔPfPRMT5 parasites were due to defects in the formation of merozoites and the invasion capacity of the merozoites. Both ΔPfPRMT5 lines showed an apparent defect in cell proliferation. ΔPfPRMT5-1 and ΔPfPRMT5-2 produced $15.51 \pm 0.31$ ($n = 197$) and $15.96 \pm 0.67$ ($n = 239$) merozoites per segmenter, respectively, which were significantly fewer than $18.01 \pm 0.82$ ($n = 238$) merozoites per segmenter produced in 3D7 (Fig. 2c, ANOVA, $P < 0.05$). In addition, we found that the invasion rate of the merozoites released from the ΔPfPRMT5−1 schizonts was approximately three times lower than that from the WT 3D7 in an in vitro merozoite invasion assay (Fig. 2d, $P < 0.01$, $n = 3$). Altogether, these data indicated that the slower in vitro growth of the ΔPfPRMT5 lines was mostly attributed to the lower numbers of merozoites formed and their reduced invasion rates.

**PfPRMT5 disruption substantially reduces H3R2me2s and overall sDMA levels**. To investigate the effect of *PfPRMT5* disruption on histone arginine methylation, we purified parasite histones at the schizont stage and determined methylation at H3R2, H3R8, and H4R3 using methylation-specific antibodies. Compared to the WT parasites, the H3R2me2s level was substantially reduced in the disruptant lines, whereas H3R8me2s and H4R3me2s did not show noticeable changes in abundance, indicating that *PfPRMT5* disruption specifically affected symmetric methylation of H3R2 (Fig. 2e). Similarly, depletion of PRMT5 in murine and human ES cells resulted in no changes of H4R3me2s and H2AR3me2s, respectively[52,53]. Analysis of asymmetric dimethylation of H3R2 showed that the H3R2me2a levels were comparable between the ΔPfPRMT5 and the WT, supporting that H3R2me2a is deposited by a Type I PRMT.

Enzymatic assays verified PfPRMT5 as a Type II PRMT that catalyzes the formation of MMA and sDMA. To evaluate whether *PfPRMT5* disruption affected the overall arginine methylation pattern in the parasite, we monitored the MMA, aDMA, and sDMA levels by Western blots using anti-MMA, anti-aDMA, and anti-sDMA antibodies, respectively (Fig. 2f). All antibodies detected abundant arginine methylation in the WT parasites with the most extensive methylation of proteins in molecular masses of 60–80 kDa. Using the same amounts of total parasite lysates from the WT parasites and the ΔPfPRMT5-1 line, we found that both anti-MMA and anti-aDMA antibodies detected protein bands of similar patterns and intensities (Fig. 2f). In contrast, the anti-sDMA antibodies revealed a substantial reduction in the number of protein bands and their intensities in the ΔPfPRMT5-1 line. This result corroborated PfPRMT5 as a

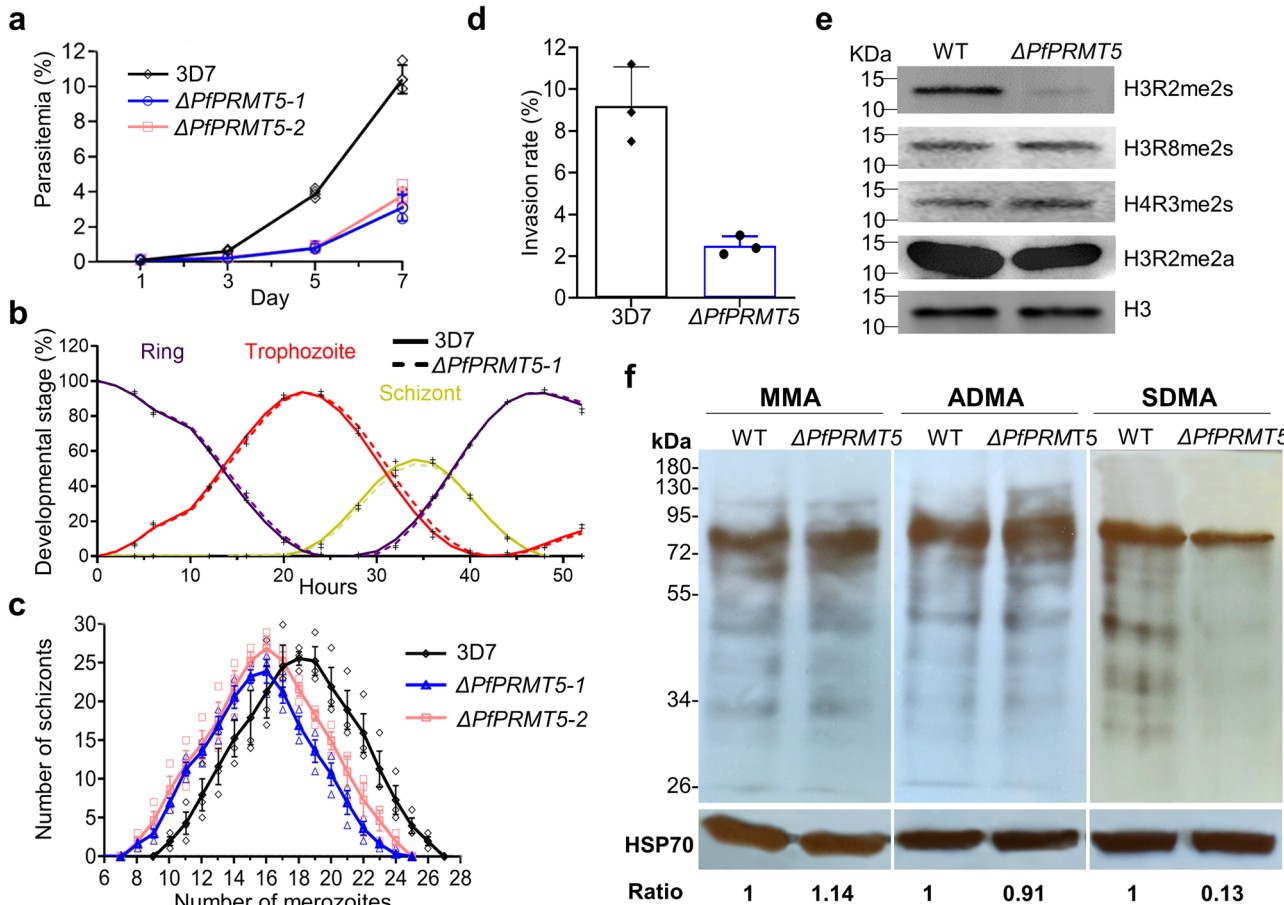

**Fig. 2 Alteration of parasite growth and protein modifications upon *PfPRMT5* disruption. a** Growth curves of two *PfPRMT5* disruptants (*ΔPfPRMT5-1* and *ΔPfPRMT5-2*) compared to WT 3D7 parasite. The disruptants grew significantly slower than 3D7 (ANOVA, P < 0.01, n = 3 biological replicates). **b** The IDC progression profiles of the *PfPRMT5* disruptant (*ΔPfPRMT5-1*) along with 3D7 control showing the proportions of the ring, trophozoite, and schizont stages through a 52 h period. The dynamics of the ring, trophozoite, and schizont stages were very similar between the 3D7 and *ΔPfPRMT5-1*. **c** Distribution of merozoite numbers in schizonts from 3D7 and two *PfPRMT5* disruptants. The *PfPRMT5* disruptants had fewer numbers of merozoites in schizont than in 3D7 (ANOVA, P < 0.05, n = 3 biologically independent experiments). **d** The capacity of invasion by merozoites from PfPRMT5 disruptant (*ΔPfPRMT5-1*). The invasion rate is significantly lower than the 3D7 control (P < 0.01, n = 3 biologically independent experiments). **e** Western blots show a substantial decrease of H3R2me2s and relatively stable H3R8me2s, H4R3me2s, and H3R2me2a in the schizonts of PfPRMT5 disruptant (*ΔPfPRMT5-1*). H3 was used as a loading control. **f** Western blots indicate the pattern of overall protein arginine methylation (MMA, aDMA, and sDMA) between *ΔPfPRMT5-1* and WT (3D7) parasites. A considerable reduction in the sDMA level was detected in *ΔPfPRMT5-1*. PfHSP70 was used as an equal loading control. The relative levels of MMA, aDMA, and sDMA were normalized with PfHSP70. The standard deviation was shown as error bars.

type II PRMT in *P. falciparum*, responsible for depositing the sDMA mark on many cellular proteins.

***PfPRMT5* disruption results in a reduced expression of invasion-related genes**. Histone arginine methylation by PRMT5 is associated with global transcriptional changes in organisms studied[22]. To explore genome-wide transcriptional changes during the IDC upon *PfPRMT5* disruption, we performed microarray analysis using parasite RNA isolated from highly synchronized cultures at 12, 24, 36, and 46 h post-invasion (hpi). Results from three biological replicates showed that the overall transcriptomes between the WT 3D7 and *ΔPfPRMT5* were similar, with linear correlation ranging from 0.97 to 0.99 (Fig. 3a, Supplementary Data 1). Yet, Significance Analysis of Microarrays (SAM) revealed drastically increased effects of *PfPRMT5* disruption on gene expression toward the later stages of the IDC, consistent with the higher abundance of PfPRMT5 protein in late trophozoites and schizonts. After excluding variant gene families showing altered expression, 635 transcripts were significantly down-regulated in the *ΔPfPRMT5* parasites (Fig. 3b, Supplementary Data 1). *ΔPfPRMT5* minimally affected

ring-stage gene expression at 12 h with only 2 down-regulated genes, whereas 110, 265, and 307 transcripts were significantly down-regulated at 24, 36, and 46 h of the IDC, respectively (Fig. 3c). Among these down-regulated genes, Gene ontology (GO) analysis identified strong enrichment of genes involved in protein export to the parasitophorous vacuole and invasion and motility-related categories at the later stages (Fig. 3d). Specifically, 70 and 30 genes down-regulated at 36 h and 46 hpi, respectively, are associated with invasion and motility (Supplementary Data 1). Moreover, almost all 86 genes encoding invasion-related proteins[14,54] were down-regulated to various extents at these two-time points (Fig. 3e, Supplementary Fig. 7). It is also noteworthy that 12 and 13 protein kinases/phosphatases, including invasion/egress-related calcium-dependent protein kinase 1 (*PfCDPK1*) and *PfPKG*[55,56], were among the down-regulated transcripts encoding enzymes at 36 and 46 hpi in *ΔPfPRMT5*, respectively (Supplementary Data 1).

**H3R2me2s is restricted to discrete chromatin domains and positively correlated with gene expression**. Earlier studies showed that PRMT5 methylates H3R2 and recruits coactivator

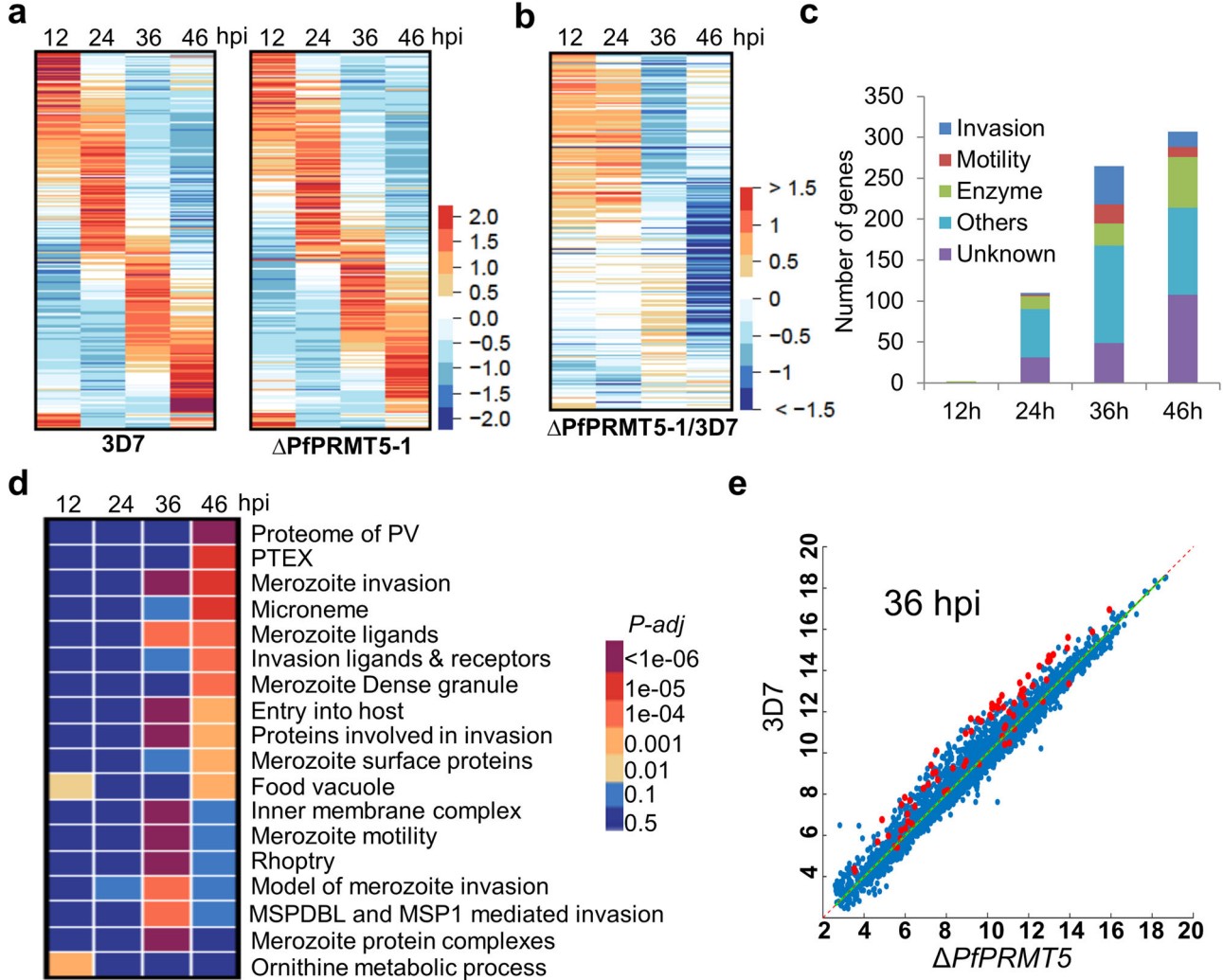

**Fig. 3 Transcriptome changes in *P. falciparum* during the IDC upon *PfPRMT* disruption. a** The phaseograms of the transcriptomes from WT 3D7 and *ΔPfPRMT5-1* show the disturbance of the cascade-like gene expression pattern in the disruptant line at four-time points of the IDC. hpi, hours post-invasion. **b** The expression fold changes of the transcript levels between ΔPfPRMT5-1 and WT from the transcripts identified by SAM at four-time points. **c** Proportions of genes by functional categories from the transcripts identified by SAM at the four-time points of the IDC. **d** GO enrichment analysis of the genes identified by SAM. **e** Comparison of transcriptomes (log2 value) between *ΔPfPRMT5-1* and WT at 36 hpi. The 86 invasion-related genes are shown as red dots.

complexes to promoters to activate the expression of target genes, which are down-regulated upon *PRMT5* disruption[23,24]. Since *PfPRMT5* disruption resulted in the down-regulation of invasion genes (Fig. 3) and the substantial reduction of the H3R2me2s occurred in schizonts (Fig. 2e), we speculated that PfPRMT5 might be involved in activating invasion genes by methylating H3R2 at their promoters. To validate this hypothesis, we determined the genome-wide distribution of the H3R2me2s mark in schizonts using the Cleavage Under Targets and Tagmentation (CUT&Tag) technique, a recently developed enzyme-tethering strategy for the high-resolution, low-background profiling of the chromatin landscape[57]. The three replicates of CUT&Tag-seq using nuclei from the schizont stage showed high reproducibility, with correlation coefficients of ~0.8. Peak calling using criteria of presence in at least two of three replicates with at least 50% overlap of the peaks identified 1849 H3R2me2s signals (Fig. 4a, Supplementary Data 2), distributed among the 14 chromosomes (Supplementary Fig. 8a). Among these peaks, 62% (1147) are localized at 5' UTR regions of 783 genes, 31.2% (577) at coding regions of 470 genes, and 6.8% (125) at 3' UTR regions of 115 genes. A similar peak distribution was also found in mammalian

cells[23,58]. The peaks are ~17.44 bp wide and ~1265 bp from the ATG sites (Supplementary Data 2).

We next wanted to determine if the H3R2me2s signals in the 5' UTRs were involved in transcriptional regulation. Comparing the transcriptome data showed that genes with H3R2me2s enrichment in the 5' UTRs had significantly higher expression levels than other genes in the genome, suggesting that H3R2me2s is associated with gene activation (Fig. 4b). In addition, the H3R2me2s peaks were highly co-localized with the active mark H3K9Ac, H2A.Z, and H3K4me3[44], further implying H3R2me2s as a euchromatin mark (Fig. 4c–e). GO enrichment analysis revealed a multitude of cellular pathways, including merozoite invasion, transcription, translation, stress response, cell cycle, schizogony, cell adhesion, exit from RBC, vesicle transport, signal transduction, DNA repair, and telomere organization, which are potentially regulated by this euchromatic mark (Fig. 4f, Supplementary Data 2). Besides the many genes directly involved in the invasion process (merozoite apical structure, tight junction, and movement), AP2-I, a major transcription factor (TF) regulating merozoite invasion[12], was among the H3R2me2s-enriched AP2-TFs (Fig. 4g, Supplementary Data 2).

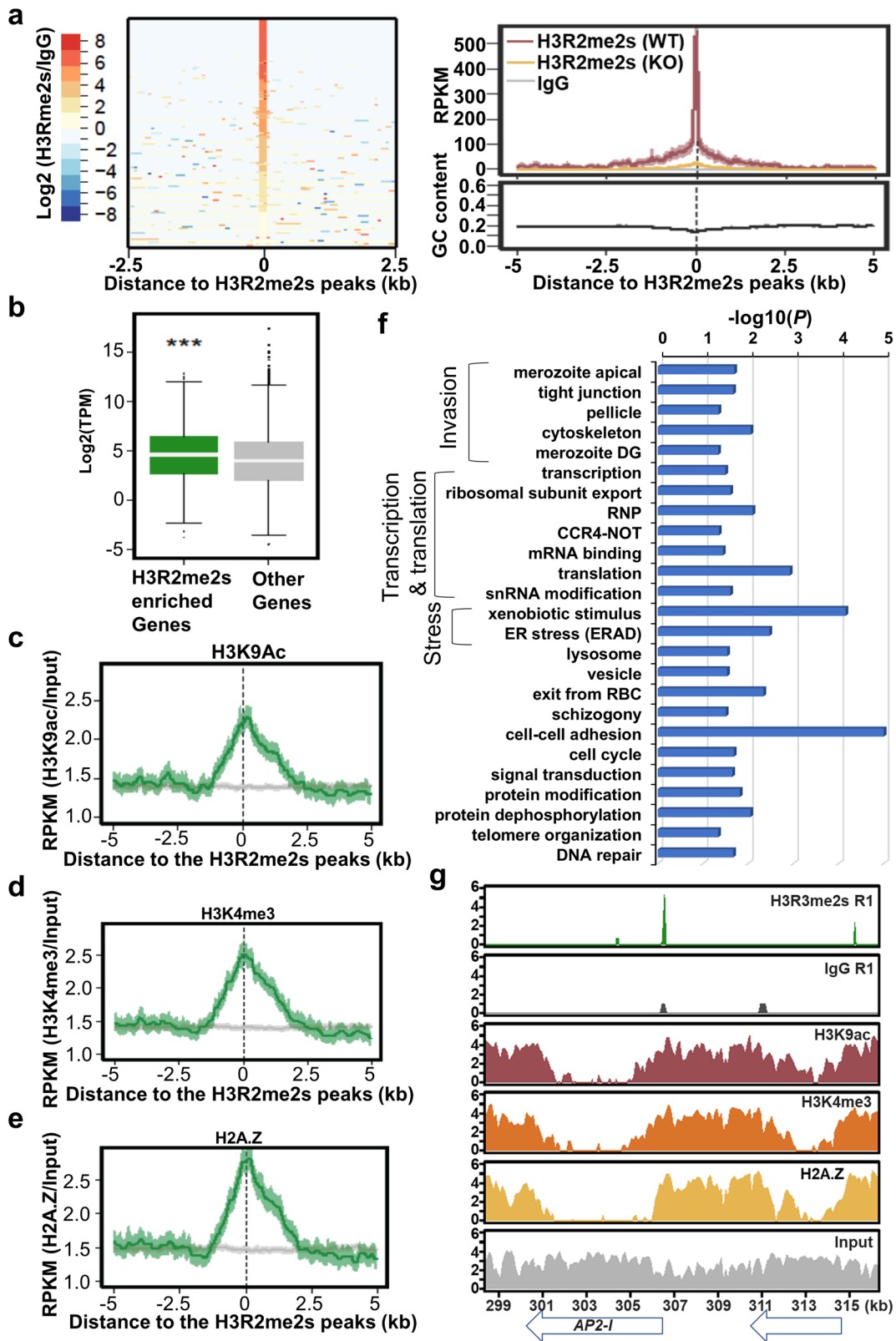

Furthermore, 110 genes are shared between the 783 genes with H3R2me2s peaks in their 5' UTRs and 540 genes downregulated at late stage upon PfPRMT5 disruption, including seven invasion genes, *PfCDPK1*, three inner member complex (IMC) associated genes (Supplementary Data 2). Collectively, CUT&Tag-seq analysis revealed that H3R2me2s regulates diverse cellular pathways including invasion.

To further confirm that PfPRMT5 primarily methylates the H3R2me2s in vivo, chromatin signals of this histone mark in the late-stage *ΔPfPRMT5* parasites were analyzed by three replicates of CUT&Tag-seq. By comparing to the IgG negative controls, no or only low numbers (17 and 12) of peaks were called from three replicates. No real peaks of H3R2me2s were identified in the *ΔPfPRMT5* based on the criteria of the presence in at least two of

**Fig. 4 H3R2me2s chromatin landscape in the schizont stage. a** A heatmap (left) of 1468 genomic loci from CUT&Tag-seq replicate 1, flanked by 2.5 kb on each side, shows the enrichment of H3R2me2s compared to IgG (log2 of RPKM H3R2me2s/RPKM IgG). The top right plot shows the strong enrichment of H3R2me2s (red) in the WT parasites, and depletion of H3R2me2s enrichment in the ΔPfPRMT5-1 (yellow) compared to the control IgG (grey) based on CUT&Tag-seq replicate 1. The increase in signal intensity is unrelated to the GC content (bottom right). **b** A box-whisker plot shows higher expression of the genes enriched with H3R2me2s at their 5′ UTRs compared to the rest of the genes. ***$P < 0.001$ (Wilcoxon test). **c–e** The H3R2me2s peaks at the 5′ UTRs were highly colocalized with the H3K9Ac, H3K4me3, and H2A.Z. The H3R2me2s CUT&Tag-seq peak profiles were aligned with the published corresponding ChIP-seq profiles. **f** GO enrichment of 791 genes enriched with H3R2me2s at the 5′ UTRs from at least two of three CUT&Tag-seq replicates. *P* values were shown after transformation by -log10. **g** Signals of H3R2me2s and IgG control in the *AP2-I* (PF3D7_1007700) locus from replicate 1 (R1) of the CUT&Tag-seq experiment, along with ChIP-seq data of H3K9Ac, H3K4me3, and H2A.Z. The signals were shown by log2 transformation of the number of reads in each peak.

three replicates with at least 50% overlap of the peaks. The complete depletion of H3R2me2s in the ΔPfPRMT5 parasite was observed at the sites of the H3R2me2s peaks in the WT (Fig. 4a).

**Coordinated role of PfPRMT5 and H3R2me2s in regulating gene expression.** We then attempted to analyze the chromatin signals of PfPRMT5 in the PfPRMT5::PTP parasite line compared to the 3D7 wildtype parasite line by using IgG because this antibody can specifically bind two IgG-binding units of protein A of *Staphylococcus aureus* (ProtA) in the PTP tag. After mapping and peak calling, we identified 334 PfPRMT5 peaks using the same criteria for peak detection (Supplementary Data 3). 78% (261) of them were localized in the 5′UTRs of 28 genes. These 28 genes consist of 23 *var* genes, dipeptidyl aminopeptidase 3 (*DPAP3*), DNA helicase PSH3, heat shock protein 70 (*HSP70x*), and acyl-CoA binding protein (isoform 2), and *PfCDPK1*. Among them, *PfCDPK1* and *DPAP3* were found significantly down-regulated after PfPRMT5 disruption in our transcriptomic study (Supplementary Data 1, 3).

Compared to 1147 H3R2me2s peaks at 5′ UTR regions of 783 genes, PfPRMT5 peaks at 5′UTRs in 27 of 28 genes, including *PfCDPK1*, are overlapped with the corresponding H3R2me2s peaks (Supplementary Data 3). The signals of the shared H3R2me2s peaks are significantly higher than the rest of the H3R2me2s peaks (Supplementary Fig. 8b, $P < 0.01$), indicating that CUT&Tag-seq of PfPRMT5 could only identify the locations with high PfPRMT5 occupancy, which potentially creates higher H3R2me2s signals in the chromatin. These results are consistent with the published studies which showed that histone modifiers are notoriously difficult to be studied by ChIP-seq because of their highly dynamic nature and relatively weak binding to the chromatin, such as GCN5[59]. Previous studies on PRMT5 and its histone marks in eukaryotic cells were conducted by ChIP-PCR of PRMT5 along with ChIP-seq of its histone marks, suggesting that ChIP-seq of PRMT5 was difficult in those studies[60,61]. Recent reports showed two separate ChIP-seq studies identified ~3000 genes whose promoters or gene bodies were enriched with PRMT5 reads in pre-B leukemia and Hela cell lines[62,63]. In contrast, ChIP-seq in chicken and pro-B cells revealed several folds more genes enriched with H3R2me2s peaks[24,58]. Taken together, there are intrinsic difficulties for PRMT5 ChIP-ing than its histone marks.

To confirm that PfPRMT5 and H3R2me2s mediate the regulation of invasion genes, we performed ChIP-qPCR to measure the enrichment of PfPRMT5 and H3R2me2s in the 5′ UTRs of the three invasion genes (*AMA1*, *EBA175*, and *MSP1*) at the late schizont stage (active stage) compared to the ring stage (silence stage). The transcription of these genes was reduced (significantly for *AMA1* and *EBA175*) after *PfPRMT5* disruption (Fig. 3e, Supplementary Data 1). Compared to the signals of PfPRMT5 and H3R2me2s in the ring stage, both signals are substantially enriched in the 5′ UTRs of the selected genes at the schizont stage whereas a control gene (*VAR2CSA*) did not show

enrichment in both stages (Fig. 5, Supplementary Fig. 8c). The fold changes of H3R2me2s signals between schizont and ring stages were in most cases higher than those of PfPRMT5, again suggesting that ChIP-ing for PfPRMT5 is more difficult than H3R2me2s. These ChIP-qPCR results are somehow in agreement with the H3R2me2s CUT&Tag-seq data. H3R2me2s signals in the 5′ UTRs of *MSP1, AMA1*, and *EBA175* were detected in all three (*MSP1*) or two replicates (*AMA1* and *EBA175*) of CUT&Tag-seq, respectively (Supplementary Fig. 9). However, only one peak for each gene was called by SEACR[64] because the read counts in some replicates were not high enough to confidently be called peaks. Because our criteria used for defining real peaks are the presence in at least two of three replicates, these single peaks were not considered real peaks.

**PfPRMT5 interactome suggests its involvement in transcriptional regulation and RNA metabolism.** We conducted immunoprecipitation (IP) using the PfPRMT5::PTP parasite line to identify the PfPRMT5 interactomes in both the cytosolic and nuclear fractions. A single-step IP with the IgG Sepharose beads was performed to capture proteins transiently or weakly associated with PfPRMT5. The bound proteins were released by TEV proteinase digestion and identified by liquid chromatography coupled with tandem mass spectrometry (LC-MS/MS). The same IP procedure was performed with the WT 3D7 as the control, and the IP data were subjected to Significance Analysis of INTeractome using stringent criteria (a probability of >95% and false discovery rate [FDR] of <1%)[65]. From two biological replicates, we identified 480 and 247 PfRPMT5-associated proteins from the cytosolic and nuclear extracts, respectively, of which 177 proteins were shared between the two fractions (Fig. 6a, Supplementary Data 4). Compared to the nuclear and chromatin-bound proteome data and integrating the protein localization information from PlasmoDB[66,67], 72% (345/480) and 75% (186/247) of the identified proteins from cytosolic and nuclear IPs were consistent with the localization in the cytosol and nucleus, respectively (Fig. 6b, Supplementary Data 4). GO enrichment analysis of the PfPRMT5-associated proteins from both compartments detected similar terms of enrichment such as RNA splicing, mRNA processing, regulation of gene expression, vesicle transport, and DNA replication. In addition, some biological processes were enriched only in one of the two fractions, e.g., decapping of mRNA, ribosome biogenesis, regulation of translation, and dephosphorylation in the cytosolic IP, and nuclear pore in the nuclear IP (Figs. 6c, 6d, Supplementary Data 4). This result indicated that PfPRMT5-associated proteins are involved in multiple cellular pathways.

The enrichment of the GO term "regulation of gene expression" is consistent with the findings from many studies showing that PRMT5 forms complexes with specific TFs or chromatin regulators to regulate transcription[18,23,24]. The PfPRMT5 interactomes include several AP2 transcription factors, PfGCN5, PfADA2, and a few WD domain-containing proteins,

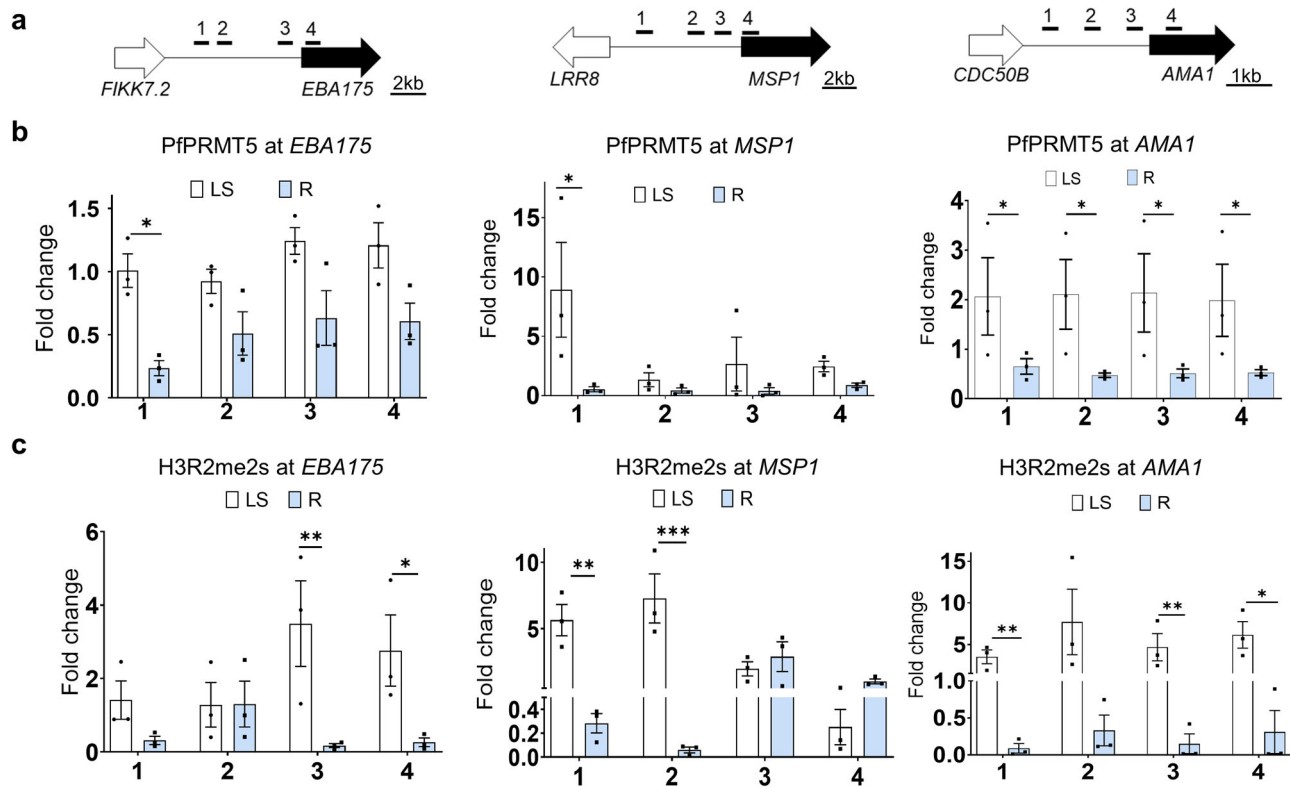

**Fig. 5 PfPRMT5 and H3R2me2s coordinately regulate invasion genes.** The enrichment of PfPRMT5 and H3R2me2s was determined at the late schizont (LS) and ring (R) stages by ChIP followed by qPCR. **a** Schematics of genomic loci and primer pairs marked as 1, 2, 3, and 4 located in the 5′ UTR regions of *EBA175*, *MSP1*, and *AMA1*. **b** ChIP-qPCR detected the enrichment of PfPRMT5 in the 5′ UTR regions of the selected genes. **c** ChIP-qPCR monitored the enrichment of H3R2me2s in the 5′ UTR regions of *EBA175*, *MSP1*, and *AMA1*. The fold change indicates the enrichment relative to the reference gene *seryl-tRNA synthetase* (PF3D7_0717700). *, **, and *** indicate $P < 0.05$, 0.01, and 0.001, respectively, Mann–Whitney U test, $n = 3$ biologically independent experiments. The standard deviation was shown as error bars.

suggesting the participation of PfPRMT5 in regulating specific genes and chromatin states (Supplementary Data 4). Two transcriptional regulators of invasion-related genes, AP2-I, and BDP1[12,14] were also present in the nuclear IPs, consistent with PfPRMT5's role in regulating merozoite invasion (Supplementary Data 4). Of note, proteins identified from the cytosol included four subunits of the signal recognition particle (SRP) for protein targeting the endoplasmic reticulum (ER) (Supplementary Data 4). Two of these SRP subunits in human cells, SRP68 and SRP72, function as transcriptional regulators, whose activities are regulated by PRMT5[68], suggesting that PfPRMT5 might play a similar role.

PfPRMT5-associated proteins in the RNA metabolic pathways were further grouped into RNA splicing, mRNA, rRNA, and tRNA processing (Supplementary Data 4). Notably, 37 and 19 proteins were identified as splicing-associated proteins from the cytosolic and nuclear IPs, including the spliceosomal core proteins SmD2, SmD3, and four Sm-like proteins (Lsm 1-4). SmD1, which was previously found to be associated with PfPRMT5[39], was identified in cytosolic IPs at an 88.7% probability and 2.57% FDR (Supplementary Data 4). Other splicing-associated proteins consist of many spliceosome subunits in the U2, U4/U6, and U5 subcomplexes and regulators of alternative splicing such as polypyrimidine tract-binding protein (PTB) and one of the serine/arginine-rich (SR) proteins, the alternative splicing factor ASF-1. Interestingly, two SR protein kinases (PfSRPK1 and PfSRPK2, also known as PfCLK1 and PfCLK2), which respond to the regulation of ASF-1 by phosphorylation[69], were also identified in the IPs. The identified proteins associated with mRNA processing include subunits of

the CCR4-NOT complex and P granule, mRNA-decapping enzymes, poly A-binding proteins, pre-mRNA 3'-end processing, mRNA polyadenylation, and nonsense-mediated decay. Proteins identified in the rRNA and tRNA processing include factors involved in ribosome biogenesis and factors in tRNA modifications. These results emphasize that PfPRMT5 is extensively involved in many aspects of RNA metabolism.

**PfPRMT5 disruption causes aberrant RNA splicing**. In mammalian cells, methylation of Sm proteins by PRMT5s is essential for their assembly into mature small nuclear ribonucleoproteins (snRNPs), and disruption of the *PRMT5* gene causes aberrant RNA splicing[25–27,70]. Besides the association between PfPRMT5 and the spliceosomal core complex member PfSmD1 in *P. falciparum*[39], many PfPRMT5-associated proteins potentially function in RNA splicing (Fig. 6, Supplementary Data 4), prompting us to determine whether PfPRMT5 is important for RNA splicing. RNA-seq analysis of strand-specific mRNA libraries from the *ΔPfPRMT5* and WT lines showed that >90% of sequencing reads mapped to the *P. falciparum* genome with high quality, resulting in nearly 40 × coverage (Supplementary Data 5). While RNA-seq analysis revealed similar differential gene expression patterns as those observed in the microarray studies (Supplementary Data 5), we focused our analysis on RNA splicing events. The modified DEXseq analysis identified that 800, 1056, 479, and 1158 alternative splicing events (alternative 5' splice site, alternative 3' splice site, retained intron, and skipped exon) were altered in the *ΔPfPRMT5* parasite line as compared to the WT at four development stages, respectively (Fig. 7a, Fig. 7b, Supplementary Data 5). We further confirmed the transcripts of eight

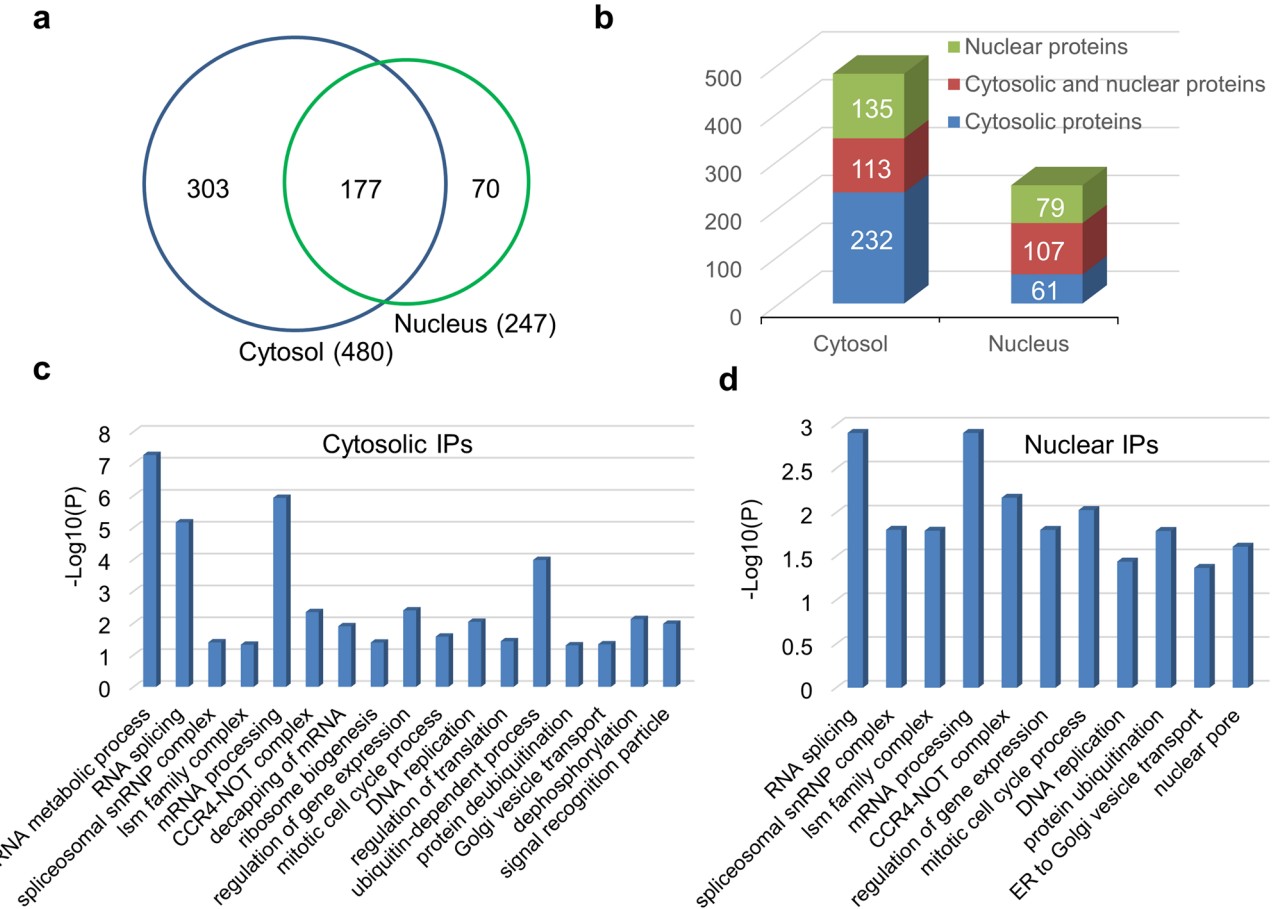

**Fig. 6 Identification of PfPRMT5-associated proteins in parasite cytosol and nucleus. a** Venn diagram shows the number of PfPRMT5-associated proteins identified by affinity purification from PfPRMT5::PTP cytosol and nucleus. **b** the allocation of identified proteins based on their cellular localization. **c** and **d** GO enrichment analysis of identified proteins from the cytosol (**c**) and nucleus (**d**).

selected alternative splicing by RT-qPCR to verify the accuracy of the RNA-seq analysis (Supplementary Fig. 10). Similar to the observation made in mammalian cells, most alterations in the ΔPfPRMT5 parasite line were the increment of splicing events, with exon skipping as the most common change[25,27] (Fig. 7a). These corresponded to 530, 600, 354, and 719 genes being impacted by these splicing events at the four development stages, respectively (Fig. 7c, Supplementary Data 5). GO enrichment analysis showed that our identified splicing events overlap with the known alternative splicing events identified by an earlier RNA-seq analysis[42] (Fig. 7d). Interestingly, genes associated with invasion, binding between host RBC and parasite, and exportome were disturbed by alternative splicing after the disruption of PfPRMT5 (Fig. 7d). Given that alternative splicing commonly results in non-translational products, an increase of alternative splicing in invasion-related genes after the disruption of PfPRMT5 may diminish the abundance of their proteins, partially contributing to the defects in the invasion.

## Discussion

We comprehensively examined the functions of PfPRMT5 by genetic disruption, transcriptome, interactome, chromatin landscape, and RNA splicing analyses and identified the conserved role of PfPRMT5 in RNA splicing and a parasite-unique role in regulating the invasion process. Consistent with the prediction based on PfPRMT5 sequences, PfPRMT5 displayed type II PRMT activity to symmetrically di-methylate histones H3 at R2 and R8,

and H4 at R3 in vitro, whereas PfPRMT5 disruption specifically reduced H3R2me2s and overall sDMA. This activity toward histones H3 and H4 is similar to the mammalian PRMT5[22], which also reflects the conservation of the N-termini of H3 and H4 between P. falciparum and humans[6]. In mammalian cells, the dimerization of PRMT5 and association with other components, such as Methylosome protein 50 (MEP50), plCln, kinase RioK1, and Grg4, are required for its catalytic activity and substrate specificity[18,19,47,48]. Human PRMT5/MEP50 methylated free H3, H4 in the H3/H4 tetramers, H3 in the mono- or oligonucleosome from Hela cells, in contrast, it failed to methylate H3 in the recombinant nucleosome[47]. Human PRMT5/Grg4 methylated both H3 and H4 in the mononucleosome whereas it only methylated H3 in the core histones[48]. We found that only PfPRMT5 purified from parasites possessed methylase activity, and it required mononucleosomes, not individual histones, as the substrates, suggesting that PfPRMT5's activity and substrate specificity on histone may also need interactions with partners. In contrast, the TbPRMT5 purified from bacteria and parasites showed efficient methyltransferase activities to free histones[34]. While our interactome studies identified potential partners of PfPRMT5, further studies are needed to determine the critical partners required for its activity.

The mammalian PRMT5 has a dual role in depositing both repressive and active histone marks, depending on what complexes PRMT5 resides. H3R8me2s and H4R3me2s are repression marks for transcription in model organisms, and PRMT5 is associated with numerous transcription factors and repressor

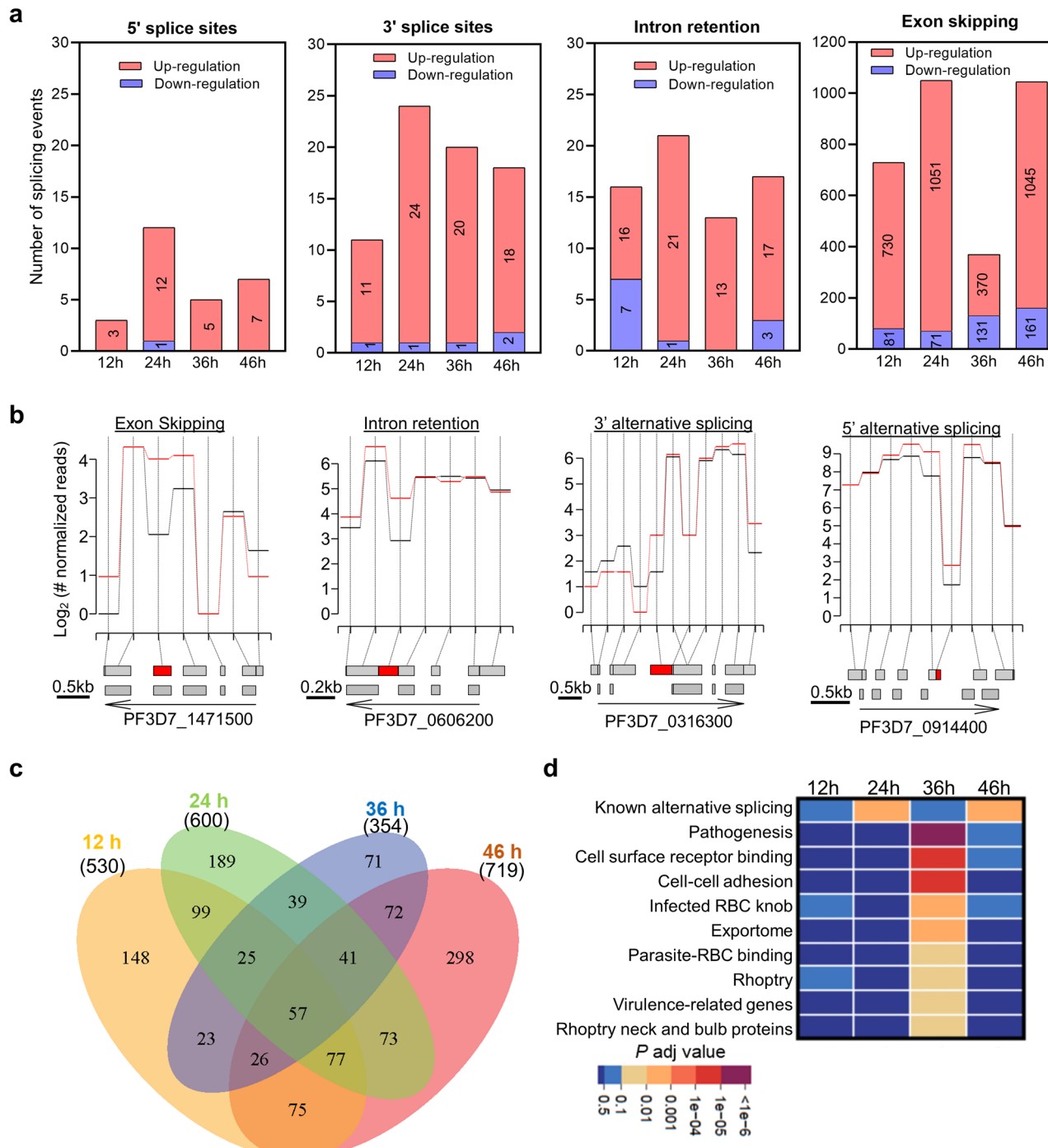

**Fig. 7 Alteration of alternative splicing upon *PfPRMT5* disruption. a** Occurrence of alternative splicing events in genes that were up-regulated (red) or down-regulated (blue) in *ΔPfPRMT5* parasites compared to the WT. The types of alternative splicing (5' splice sites, 3' splice sites, intron retention, and exon skipping) are shown as different bar graphs. **b** Examples of the four types of alternative splicing events. The annotated exon/intron from the known genome model and transcriptional bins identified from RNA-seq were depicted at the bottom of each diagram with a red box denoting altered alternative splicing events. The expression levels of each transcriptional bin in *ΔPfPRMT5* and WT parasite are shown as the red and black curves in the diagram, respectively. **c** Venn diagram showing the number of genes with changed alternative splicing at the four IDC stages. **d** GO term enrichment analysis of the genes with changed alternative splicing at the four IDC stages.

complexes[22,71]. In contrast, PRMT5 is also a coactivator since H3R2me2s recruits coactivator complexes and is highly correlated with the active mark H3K4me3 at active promoters[23,24,47,72]. In our study, although we found that PfPRMT5 could deposit the putative repressive (H3R8me2s and H4R3me2s) and the active (H3R2me2s) histone marks in vitro, we only found evidence of a substantial decrease of the H3R2me2s mark after *PfPRMT5*

disruption. In agreement, *PfPRMT5* disruption predominantly led to the down-regulation of gene expression, suggesting PfPRMT5 mainly serves as a coactivator during the *P. falciparum* IDC. In comparison, disruption of PRMT5 in model organisms normally causes a higher number of up-regulated transcripts than down-regulated ones[25–27], perhaps due to the de-repression of PRMT5-regulated transcriptions. Furthermore, our analysis of the

chromatin landscape by CUT&Tag-seq further confirmed that H3R2me2s is an active mark, overlapping significantly with other euchromatin marks H3K9ac, H3K4me3, and H2A.Z. This active mark is enriched in genes involved in invasion, translation, stress response, cell cycle, cell adhesion, signal transduction, DNA repair, and telomere organization, suggesting PfPRMT5's participation in transcriptional regulation of a multitude of cellular processes, which deserves further in-depth elucidation.

Although CUT&Tag-seq only revealed that PfPRMT5 occupied a fraction of H3R2me2s regulated genes due to the intrinsic difficulties, our ChIP-qPCR analysis demonstrated the coordinated role of PfPRMT5 and H3R2me2s in regulating invasion-related genes. Higher enrichments of PRMT5 and H3R2mes2 in the 5' UTRs of certain *var* genes suggest their potential involvement in regulating these genes in the subtelomeric regions. Although we noticed that *var* genes were substantially changed from our microarray, RNA-seq, and CUT&Tag-seq data, these data cannot be used for further in-depth analysis because our parasites are mixed clones expressing different *var* genes due to long-term culture. To further study the function of PfPRMT5 and H3R2me2s in the regulation of these important genes, additional cloning steps are needed.

The reasons why disruption of *PfPRMT5* only resulted in the substantial decrease of H3R2me2s but no noticeable changes on H3R8me2s and H4R3me2s are probably because that silence marks (H3R8me2s and H4R3me2s) and active mark (H3R2me2s) are deposited by different PRMTs. Our Western blots showed that silence marks (H3R8me2s and H4R3me2s) consistently appeared during asexual development, in contrast, H3R2me3 only appeared at a high level in the schizont stage, consistent with the expression of PfPRMT5 (Fig. 1a), indicating that these marks are regulated differently. Additionally, disruption of PfPRMT5 only caused the reduction of the active mark (H3R2me2s), suggesting that only PfPRMT5 modifies H3R2 while another unknown PRMT modifies the silence marks (H3R8me2s and H4R3me2s). Similarly, PRMT5 purified from murine embryonic stem (ES) cells could methylate histone H2A and H4 in vitro. However, there was no significant loss of H4R3me2s modification but a selective loss of H2AR3me2s modification after the depletion of PRMT5 in murine ES cells[52]. Another report showed that the depletion of PRMT5 did not result in a change in H2AR3me2s in human ES cells[53]. Human/murine H2A and H4 contain the same R3 motif but malaria parasite H2A does not contain the R3 motif. Furthermore, our PfPRMT5 interactome study revealed that PfPRMT5 interacts with GCN5, ADA2, and several WD-domain-containing proteins, indicating PfPRMT5 is likely involved in gene activation, in agreement with the function of H3R2me2s in gene activation and the findings that H3R2me2s preferably bind a WD domain protein (WDR5) for gene activation[23]. In agreement with the above findings, H3R2me2s signals in the ΔPfPRMT5 were completely depleted in the chromatin (Fig. 4a). Although in vitro methylation assays showed PfPRMT5 could methylate H3R2, H3R8, and H4R3, they might not reflect the in vivo function of PfPRMT5. If PfPRMT5 can methylate H3R8 and H4R2 in vivo, there might be other PRMTs that can compensate for the loss of H3R8me2s and H4R3me2s by disruption of PfPRMT5.

Phenotype analysis identified defects in merozoite invasion upon *PfPRMT5* disruption, which may reflect a collective result of multiple layers of dysfunctions. While *PfPRMT5* disruption decreased the levels of H3R2me2s at the promoters of invasion-related genes, it may also affect other euchromatin marks such as H3K9ac, since the acetylation writer GCN5 and reader BDP1 were found in the PfPRMT5 interactome[13,14]. Targeting of PfPRMT5 to invasion-related genes may be achieved through its interaction with specific TFs such as AP2-I[12]. Furthermore,

elevated alternative splicing events in invasion-related genes upon PfPRMT5 disruption may also affect the stability of these mRNAs and their translation.

Unlike human PRMT5 and TbPRMT5, which localize almost exclusively to the cytoplasm[34,73], PfPRMT5 is approximately evenly distributed in the cytoplasm and nucleus. Interactome analysis with the PfPRMT5::PTP parasites revealed 480 and 247 proteins as PfRPMT5-associated proteins from the cytosolic and nuclear extracts, respectively, which may be integral components of PfPRMT5 complexes and substrates (Fig. 6). These proteins can be categorized into several functional groups, including RNA splicing, transcription, DNA replication, and translation, suggesting PfPRMT5's participation in these cellular processes. PfPRMT5's function in chromatin regulation and transcription was illustrated by its interaction with proteins involved in chromatin biology (e.g., BDP1 and GCN5) and transcriptional disturbance after *PfPRMT5* disruption. Its interactions with proteins involved in RNA metabolism and the spliceosome[74] are associated with substantial defects in RNA splicing upon *PfPRMT5* disruption. It is also expected that many of these proteins would be the substrates of PfPRMT5 given the increasing number of proteins identified in model organisms as the substrates (Supplementary Data 6)[75–77].

This study demonstrated that PfPRMT5 is involved in RNA splicing processes like PRMT5 in mammalian cells[25–27,70]. However, PfPRMT5 may function differently in the splicing process because *Plasmodium* core spliceosomal Sm proteins, which were associated with PfPRMT5, lack the RG motifs that are normally methylated by mammalian PRMT5[78], suggesting that PfPRMT5 may methylate Sm protein differently. Probably this different methylation pattern specifies the splicing process to the transcripts related to *Plasmodium*-specific pathways such as invasion.

Here we provided multiple lines of evidence showing that the type II enzyme PfPRMT5 plays critical roles in regulating general and parasite-specific biology in *P. falciparum*. The active chromatin mark H3R2me2s deposited by PfPRMT5 shares extensive overlaps with other euchromatin marks, demonstrating a high level of crosstalk among these epigenetic factors. Its specific regulation of invasion-related genes may be dictated by multiple layers of components including the chromatin structure affecting transcription, specific recruitment by TFs, and transcript stability and translation influenced by RNA splicing. The extensive interactomes of PfPRMT5 imply its involvement in multiple cellular pathways, which awaits future investigations. A better understanding of arginine methylation will enable the design of malaria therapeutics targeting this family of enzymes.

## Material and methods

**Sequence analysis and structure prediction**. Four known PRMT5 proteins from Homo sapiens (HsPRMT5/JBP1), Saccharomyces pombe (SpSKB1), Caenorhabditis elegans (CePRMT5), and Trypanosoma brucei (TbPRMT5) were retrieved from GenBank (accession numbers. O14744, P78963, NP_498112.1, and Q38CH6, respectively) and aligned with the putative PfPRMT5 (Pf3D7_1361000) sequence by Clustal Omega. The structure of PfPRMT5 was predicted by using the multiple threading alignments in I-TASSER[79], based on the crystal structure of HsPRMT5 (PDB code 4GQB)[80].

**Parasite culture**. The 3D7 clone of *P. falciparum* was cultured by the established methods with some modifications[81]. Synchronization of asexual stages was performed by two rounds of sorbitol treatment at the ring stage. For time-course studies, schizonts were purified by Percoll gradient centrifugation and mixed with fresh RBCs[82], and parasites were harvested at 12, 24, 36, and 46 h

later to represent ring, early trophozoite, late trophozoite, and schizont stages, respectively.

**Rapid amplification of cDNA ends and qRT-PCR.** Total RNA was isolated from the parasites using Trizol (Invitrogen) and treated with ezDNase enzyme (Invitrogen) to remove contaminating genomic DNA. The 5' and 3' UTRs of the *PfPRMT5* mRNA were determined by using the FirstChoice RLM-RACE (RNA ligase-mediated rapid amplification of cDNA ends) kit (Ambion) with *PfPRMT5*-specific primers 5RACE-R1, 5RACE-R2, 3RACE-F1, and 3RACE-F2 (Supplementary Data 7) as described earlier[83]. PCR products were cloned in TOPO® cloning vector (Invitrogen) and sequenced.

The relative transcription levels of *PfPRMT5* were studied by quantitative reverse transcriptase-PRC (qRT-PCR) analysis at four stages of the IDC. HotStart-IT SYBR Green One-Step qRT-PCR Master Mix Kit (USB) was used for cDNA synthesis and PCR amplification with primers PfPRMT1-F3 and PfPRMT1-R3 (Supplementary Data 7). The *seryl-tRNA synthetase* gene (PF3D7_0717700) was used as the internal reference. The expression was analyzed as described previously[6].

To validate the RNA splicing events, qRT-PCR analysis was performed with eight pairs of primers to measure the levels of transcripts from different splicing events in eight selected genes (Supplementary Fig. 9, Supplementary Data 7). Relative expression levels of alternatively spliced transcripts were determined using the $2^{-\Delta\Delta Ct}$ method with the PF3D7_0713400 (an unchanged region) as the internal reference.[84]

**Genetic manipulation of the *PfPRMT5* gene.** To tag the C-terminus of PfPRMT5 with the PTP tag[13,85,86], a *PfPRMT5* fragment [nucleotides (nt) 1138-2172] was amplified using primers F1 and R1 (Supplementary Data 7) from *P. falciparum* genomic DNA and cloned into modified pBluescript SK to fuse with the PTP and pDT 3' UTR as described earlier[37,87]. This cassette was subcloned into pHD22Y at *Bam*HI and *Not*I sites to produce pHD22Y/PRMT5-PTP[88]. To disrupt *PfPRMT5*, a *PfPRMT5* fragment of nt 52-1214 was amplified using primers F2 and R2 (Supplementary Data 7), cloned into modified pBluescript SK, and subcloned into pHD22Y at the *Bam*HI and *Not*I sites to produce pHD22Y/ΔPRMT5. Both plasmids were deposited in Addgene with the assigned ID numbers (203160 and 203161).

Parasite transfection was done following an RBC loading method[89]. The selection was performed with 2.5 nM of WR99210 for approximately four weeks until resistant parasites appeared[90]. Single clones of parasites were obtained by limiting dilution[91]. Correct integration of pHD22Y/PfPRMT5-PTP at the *PfPRMT5* locus was screened by integration-specific PCR with primers IntF and IntR (Supplementary Fig. 3, Supplementary Data 7). Correct integration of pHD22Y/ΔPfPRMT5 was confirmed by Southern blot with a specific probe located in the homologous region, which was generated by DIG-labeled PCR with primers SbF and SbR (Supplementary Fig. 6, Supplementary Data 7).

**TAP, single-bead pulldown, and mass spectrometry.** TAP of PfPRMT5 was performed using the PfPRMT5::PTP parasite line by established methods[13,85–87]. About $1 \times 10^9$ synchronized schizonts were lysed in 5 volumes of the PA150 buffer. The lysate was centrifuged for 10 min at $16,000 \times g$, and the supernatant was incubated with 100 μl of IgG agarose beads (GE Healthcare) at 4 °C for 2 h. The beads were equilibrated with the TEV buffer. 2 ml of TEV buffer and 150 U of TEV protease were added to the beads, and the column was rotated overnight at 4 °C. The eluate was collected and incubated with anti-protein C beads for 2 h at 4 °C. The beads were washed, and binding proteins were eluted

with an elution buffer. The final eluate was collected for in vitro methylation assay.

The first affinity purification of TAP was used to capture proteins transiently or weakly associated with PfPRMT5. After incubating the cytosolic or nuclear lysate with the IgG Sepharose beads followed by digestion with TEV protease, the eluate was separated in a 10% Bis-Tris SDS-PAGE gel for 10 min. Proteins in gel were excised, and in-gel digestion was done as described[92]. The digests were analyzed by LC/MS/MS using a Waters NanoAcquity HPLC system interfaced with a Q Exactive™ Hybrid Quadrupole-Orbitrap Mass Spectrometer (Thermo Scientific). Peptides were loaded on a trapping column and eluted over a 75 μm analytical column at 350 nL/min. MS and MS/MS were performed at 70,000 FWHM and 17,500 FWHM resolutions, respectively. The fifteen most abundant ions were selected for MS/MS. Parasite proteins were identified by searching the UniProt *P. falciparum* protein database (v01/2014). Data were filtered at 0.7–1% protein and 0.2% peptide FDR, and at least two unique peptides per protein. Mascot DAT files were parsed into the Scaffold software for validation and filtering to create a non-redundant list per sample.

**In vitro methylation assay.** PfPRMT5-PTP was purified from $1 \times 10^8$ schizonts of the PfPRMT5::PTP parasite line. In vitro PfPRMT5 activity was determined using a methylation assay as described[37]. Briefly, 2 μg of bovine core histones (H2A, H2B, H3, and H4) (Sigma), or purified *P. falciparum* core histones (H2A, H2B, H3, and H4)[87,93,94], or human mononucleosome assembled from recombinant human histones without any PTM (EpiCypher #SKU: 16-0009) were incubated at 30 °C for 2 h within 20 μl of methylation assay buffer containing 1 μCi of [³H] S-adenosylmethionine[37,87]. The reactions were resolved by 15% SDS/PAGE for fluorography using an auto-radiographic enhancer (Perkin Elmer).

**Dot blot analysis.** Dot blot was used to verify the specificity of the H3R2me2s antibodies (Supplementary Fig. 4). The modified histone H3 peptide (EpiCypher #SKU 12-0235 H3R2me2s, biotinylated) and the unmodified *P. falciparum* histone H3 peptide (250 ng of each peptide) were spotted on a PVDF membrane followed by blocking with 1% casein in Tris-buffered saline (TBS) and incubating with H3R2me2s and H3 antibodies at 1:2000 dilutions. After washing with TBS, the blot was incubated with horseradish peroxidase (HRP)-conjugated goat anti-rabbit IgG secondary antibody diluted at 1:5000 in 1% casein-TBS. Densitometry was used to quantify the degree to which the antibodies detected each peptide. To exclude the cross-reaction of H3R2me2a antibodies to histone H3 peptide with or without H2R2me2s modification, the same dot blots were performed with anti-H3R2me2a.

**Western blots.** To study PfPRMT5 protein expression during the IDC, synchronized parasites with PTP-tagged endogenous PfPRMT5 were lysed by sonication. Equal amounts of the parasite lysates (30 μg) at each developmental stage were separated by 10% SDS/PAGE and transferred to nitrocellulose membranes. Western blotting was performed using a standard procedure with rabbit anti-protein C antibodies (1 μg/ml) as the primary and HRP-conjugated goat anti-rabbit IgG (diluted 1:3000) as the secondary antibodies. Anti-PfHSP70 antibodies were used as the loading control. The results were visualized with the ECL detection system (GE Healthcare). The grey values of the bands detected by Western blot were quantified using the ImageJ software.

To estimate the distribution of PfPRMT5 in the cytoplasmic and nuclear compartments of the parasite, ~100 μl of parasite pellet was resuspended in 300 μl of a hypotonic buffer A [10 mM HEPES, pH 7.9, 1.5 mM MgCl₂, 10 mM KCl, 0.5 mM DTT,

0.5 mM EDTA and 1% (v/v) protease-inhibitor cocktail (Roche)] and incubated on ice for 10 min. The parasites were mechanically lysed by 40 strokes in a Dounce homogenizer with a loose pestle and then centrifuged at 700 g for 20 min at 4 °C. The supernatant was centrifuged at 10,000 g for 10 min to obtain the cytoplasmic extract. The pellet was resuspended in three volumes of buffer B (20 mM HEPES, pH 7.9, 20% glycerol, 200 mM KCl, 0.5 mM DTT, 0.5 mM EDTA, 0.5% NP40, and protease inhibitor cocktail) and homogenized with a tight pestle for 40 strokes. The homogenate was centrifuged at 10,000 g for 10 min to obtain the nuclear extract. Equal ratios of the protein extracts (20 μl) were resolved on a 15% SDS-PAGE gel and detected by immunoblotting using the rabbit anti-protein C antibodies (1 μg/ml). For cytoplasmic and nuclear controls, a rabbit antiserum against the recombinant His-tagged P. falciparum histone-like protein (PF3D7_0904700) and anti-histone H3 antibodies (1:1000 dilution; Millipore) were used, respectively[37].

To determine methylation of specific histone proteins, 1.5 μg of purified PfPRMT5-PTP and 1.5 μg the human mononucleosomes (EpiCypher #SKU: 16-0009) were used in the in vitro methylation assay described above. After the reaction, proteins were resolved by SDS-PAGE and transferred to PVDF membranes for immunoblot analysis using the following primary antibodies (1:2000 dilution): H3R2me2s (Epigentek #A-3705-050), H3R8me2s (BioVision #A2046-100) H4R3me2s (Millipore #07-947), H3 (Millipore #06-755) and H4 (Millipore #04-858). The latter two were used as loading controls.

To investigate the levels of H3R2me2s, H3R8me2s, and H4R3me2s during the IDC, histones were purified from the 3D7 parasite at the ring, trophozoite, and schizont stages. Western blotting was performed using the above-mentioned H3R2me2s, H3R8me2s, H4R3me2s, H3, and H3R2me2a (Epigentek #A-3714-050, 1:2000).

To determine the changes of the MMA, aDMA, and sDMA in the WT and ΔPfPRMT5 parasites, equal amounts of parasite extracts from different parasite lines were separated in a 12% SDS-PAGE gel. Western blots were performed with anti-MMA, -aDMA, and -sDMA antibodies (Cell Signaling Technology, Inc.) as the primary antibodies. Anti-PfHSP70 antibodies were used as the loading control.

**Immunofluorescence assay (IFA).** To determine the subcellular location of PfPRMT5 and the histone marks H3R2me2s, H3R8me2s, and H4R3me2s during IDC, IFA was performed as described[95]. The PfPRMT5::PTP and WT 3D7 infected RBCs (iRBCs) were fixed with 4% (v/v) paraformaldehyde and 0.0075% (v/v) glutaraldehyde followed by permeabilization with 0.5% (v/v) Triton X-100 and blocking in 3% (v/v) BSA. Anti-protein C antibodies (1 μg/ml) (for the PfPRMT5::PTP parasite), and the H3R2me2s (1:500), H3R8me2s (1:250), and H4R3me2s (1:500) antibodies (for the WT parasite) were used as primary antibodies and FITC- and Alexa fluor 488-conjugated anti-rabbit IgG antibodies (Thermo Fisher Scientific #A27034 or #A32754) were used as secondary antibodies containing 4',6-diamidino-2-phenylindole (DAPI, Invitrogen) for nuclear staining. Images were captured using a Nikon Eclipse E600 epifluorescent microscope and processed by Adobe Photoshop CS (Adobe Systems Inc. San Jose, CA).

**Phenotype analysis.** The growth phenotype of PfPRMT5 disruption lines during the IDC was compared with the WT 3D7 as described[13,87]. To measure parasite proliferation, cultures of synchronized parasites were initiated with 0.1% rings, and parasitemia was monitored daily for seven days. Progression of parasites through the IDC was monitored using Giemsa-stained smears every 2 h. Cycle time was determined as the duration

between the peak ring parasitemias of two consecutive cycles. The number of merozoites produced per schizont was determined in mature segmenters. The nuclei in schizonts were counter-stained with Hoechst 33342 (20 mM) for 5 min, and the smears were observed by light and fluorescence microscopy. Three independent biological replications were done for each parasite line. Merozoite invasion assay was performed as described[96]. The same numbers of purified viable merozoites ($2 \times 10^6$) from the WT and ΔPfPRMT5 lines were mixed with fresh RBCs ($1 \times 10^7$) for invasion, and 24 h later, the parasitemia of culture was determined. The invasion rate was calculated as the percentage of merozoites that invaded RBCs.

**Parasite transcriptome analysis.** To compare the transcriptomes during the IDC between ΔPfPRMT5 and 3D7, we used a custom-designed expression microarray from Roche NimbleGen based on the P. falciparum 3D7 genomic sequence[83]. Total RNA was extracted with Trizol Reagent from $1 \times 10^8$ highly synchronized parasites at 12, 24, 36, and 46 h, respectively. RNA was amplified and labeled with Cy5 or Cy3 using the Amino Allyl MessageAmp II aRNA Amplification kit (Ambion, Austin). RNA labeling, array hybridization, and signal normalization were done as described[83]. The experiment was replicated three times. The phaseograms of transcriptomes were built using the established methods[13]. The differential expression was analyzed by the SAM method[97]. The functional enrichment of differential transcripts was analyzed by GO enrichment[98].

For RNA-seq, sequencing libraries were constructed from parasite RNA using the KAPA Stranded mRNA Seq kit for the Illumina sequencing platform according to the manufacturer's protocol (KAPA biosystems). Libraries were sequenced on an Illumina HiSeq 2500 in Rapid Run mode using 100 nt single read sequencing. Reads from Illumina sequencing were mapped to the P. falciparum genome sequence (Genedb v3.1) using HISAT2[99]. The coverage was analyzed by using the bedtools[100]. The expression levels and the differential expression were calculated by FeatureCounts and DESeq1[101,102].

**RNA splicing analysis.** We developed a pipeline to systemically annotate alternative splicing events in P. falciparum genome from RNA-Seq data based on published methods[103–105]. First, all annotated exons were extracted and used to compare with all transcripts from the RNA-seq data. This step was achieved by merging all Cufflinks assemblies via Cuffmerge[106], followed by incorporating the merged file into GFF annotation from P. falciparum genome (PlasmoDB version 39.0) using a custom python script, 'geneInfPick.py'. If the boundary of an exon annotated from RNA-seq data did not match the respective exon annotated in GFF, this exon was then cut into two or multiple segments (transcriptional bins) and classified as the alternative splicing events (Supplementary Fig. 11a). Second, the differential expression of these alternatively spliced transcripts between 3D7 and ΔPfPRMT5 was analyzed by DEXseq and DEseq[107,108]. To do this, the output GTF file from the first step was converted to the GFF format using python script 'dexseq_prepare_annotation.py' from package 'DEXseq', and the expression level of each transcript (number of reads) was measured by the python script 'dexseq_count.py'. DEseq was used to normalize read counts for each pair of comparisons between ΔPfPRMT5 and WT in all four-time points. The differentially expressed transcripts were identified by $\log_2$(fold change) of higher than one and expression levels in the top 75%. The output file contains four columns with transcript ID, normalized read counts in WT and ΔPfPRMT5, and a class label indicating whether the alternative splicing transcript is up- or down-regulated. Finally, a custom python

script" alternativeSplicing_annotaiton.py" was used to assign the type of alternative splicing events for each transcriptional bin (Supplementary Fig. 11b–d), and a python script 'merge.ex-on.slice.together.py' was designed to trim the false alternative splicing events that were similarly differentially expressed in both alternative transcriptional bins and the annotated exons between ΔPfPRMT5 and WT.

**Cleavage Under Targets and Tagmentation (CUT&Tag).** To determine the chromosomal occupancy of H3R2me2s and PfPRMT5, CUT&Tag was performed as described with modifications[57]. Briefly, synchronized WT 3D7 and ΔPfPRMT5-1 at 40–46 hpi schizonts were harvested and fixed with 1% for-maldehyde followed by saponin lysis. The parasite pellet was suspended in nuclear extraction buffer. Ten μl of activated Concanavalin A-coated magnetic beads (EpiCypher # 21-1401) were added to each sample (∼ $0.5 \times 10^6$ nuclei). For H3R3me2s CUT&Tag-seq, the bead-bound nuclei were resuspended in 50 μl Antibody150 buffer containing 0.5 μg of rabbit anti-H3R2me2s (Epigentek #A-3705-050) as the primary antibody and rabbit IgG (Cell Signaling Technology #2729 S) serving as a control. For PfPRMT5 CUT&Tag-seq, the bead-bound PfPRMT5::PTP nuclei were incubated with IgG while WT 3D7 nuclei were used as control. After the primary antibody was removed, nuclei were incubated with 0.5 μg of the secondary antibodies, mouse anti-rabbit IgG (Invitrogen #31213), in 50 μl of Digitonin150 buffer (Antibody150, no EDTA). After unbound antibodies were removed, the nuclei were resuspended in 50 μl Digitonin300 Buffer (Digitonin150 but with 300 mM NaCl) with 2.5 μl of CUTANA pAG-Tn5 (EpiCypher #15-1017) at room temperature for 1 h and then the nuclei were washed to remove the unbound pA-Tn5. Next, the nuclei were resuspended in 50 μl Tagmenta-tion buffer (10 mM $MgCl_2$ in Digitonin300) at 37 °C for 1 h. The beads were washed with 50 μl TAPS buffer, mixed with 5 μL SDS release buffer to quench tagmentation, and then incubated at 58 °C to release tagmented chromatin into the solution. Finally, 15 μl of 0.67% Triton-X was added to each reaction to quench SDS. PCR with appropriate barcoded primers was performed using the CUTANA High Fidelity 2x PCR Master Mix (EpiCy-pher #15-1018) according to the manufacturer's recommenda-tions. Amplified DNA libraries were captured by incubating with 1.3 × Kapa pure beads (Roche #KK8001) according to the man-ufacturer's recommendations, and 150 bp paired-end sequencing was performed on the NextSeq 550 platform.

The reads of CUT&Tag-seq from three biological replicates were mapped to the *P. falciparum* genome (PlasmoDB Release 39.0)[109] by using BWA after FastQC quality control[110]. The MACS2 version 2.2.7.1 with parameters 'pileup–extsize 10' was used to generate bedgraph file. The peak calling was finished by SEACR with parameters 'norm' and 'stringent'[64,111]. To further strengthen the reproducibility, we only kept the peaks presenting in at least two biological replicates and at least 50% overlapping for downstream analysis. Our analysis concentrated on the binding events at the promoter regions with a distance to the gene ATG from 1 bp to 2 kb. The published ChIP-seq profiles were used to interrogate the association between euchromatin markers and H3R2me3[44].

**Chromatin immunoprecipitation and quantitative PCR (ChIP-qPCR).** ChIP-qPCR was performed as described with some modifications[13]. Synchronized WT 3D7 and PfPRMT5::PTP parasite lines at the ring stage [12–16 h post-invasion (hpi), ∼5 × $10^9$ iRBCs] and late schizont stage (44–46 hpi, ∼1.5 × $10^9$ iRBCs) were harvested and crosslinked with 1% paraformalde-hyde and then neutralized by glycine (0.125 M). The fixed iRBCs were lysed with saponin (0.06% final concentration) and parasites

were treated with a lysis buffer (10 mM KCl, 0.1 mM EDTA, 0.1 mM EGTA, 1 mM DTT, 10 mM Hepes, pH 7.9, 1× protease inhibitor) and then were gently homogenized using a douncer to free the nuclei. Pelleted nuclei were sonicated in a shearing buffer (0.1% SDS, 5 mM EDTA, 50 mM Tris-HCl, pH 8.1, 1× protease inhibitor). using a rod bioruptor (Microson ultrasonic cell dis-ruptor, Misonix, Inc. USA) at high power for 20 cycles of 30 s ON/30 s OFF, resulting in sheared chromatin of approximately 100–1000 bps. Fifty μl of input samples were set aside, and the remaining chromatin was diluted in an incubation buffer (0.01% SDS, 1.5% Triton X-100, 0.5 mM EDTA, 200 mM NaCl, 5 mM Tris-HCl, pH 8.1). The chromatin (75 μl/400 ng) was incubated with rabbit anti-H3R2me2s antibodies (Epigentek #A-3705-050) followed by the addition of 20 μl of agarose beads for WT 3D7 and IgG Sepharose matrix (Cytiva 17-0969-01) for PfPRMT5::PTP. Beads were then washed with the following: buffer 1 (0.1% SDS, 1% Triton X-100, 150 mM NaCl, 2 mM EDTA, 20 mM Tris HCl, pH 8.1); buffer 2 (0.1% SDS, 1% Triton X-100, 500 mM NaCl, 2 mM EDTA, 20 mM Tris HCl, pH 8.1), buffer 3 (250 mM LiCl, 1% NP-40, 1% Na-deoxycholate, 1 mM EDTA, 10 mM Tris HCl, pH 8.1) and finally twice with buffer 4 (10 mM EDTA, 10 mM Tris HCl, pH 8). The immunoprecipi-tated (IPed) chromatin was eluted with the elution buffer (1% SDS, 0.1 M $NaHCO_3$) at room temperature. The eluted chromatin and input samples were reverse crosslinked and purified by the phenol:chloroform method. For qPCR, 10 ng per well in triplicate was used with the FastStart Universal SYBR Green Master [Rox] (Sigma-Aldrich, USA). Primer pairs targeting 5' UTRs of the selected genes were designed to amplify fragments less than 300 bp (Supplementary Data 7). Fold enrichment relative to constitutively expressed reference gene *seryl-tRNA synthetase* was calculated using the $2^{-\Delta\Delta Ct}$ method. The fold changes of binding enrichment were calculated using a formula: $2^{-[(IP\ Ct\text{-}target\ -IP\ Ct\text{-}stRNA)\ -\ (Input\ Ct\text{-}target\text{-}Input\ Ct\text{-}stRNA)]}$ for each primer set targeting specific promoter regions.

**Statistics and reproducibility.** For most experiments, three independent biological replicates were performed. The results are presented as mean ± standard deviation (SD). Results are regar-ded as significant if $p < 0.05$ as established by ANOVA. To ana-lyze the schizont numbers containing different numbers of merozoites, a $\chi^2$ goodness of fit test was first used to evaluate if the number of schizonts containing a certain number of mer-ozoites was independent of the parasite lines. Then the propor-tions of schizonts with a certain number of merozoites were compared among these cell lines based on ANOVA for each merozoite number. The ChIP-qPCR data were analyzed by Mann–Whitney U test. The comparison of expression between H3R2me2s enriched genes and the other genes was analyzed by the Wilcoxon test. To make sure the reproducibility of the experiments, three biologically independent experiments were conducted for growth phenotype (growth curve, merozoite number, invasion rate), RT-PCR, and ChIP-qPCR.

**Reporting summary**. Further information on research design is available in the Nature Portfolio Reporting Summary linked to this article.

## Data availability

Microarray, RNA-Seq, and CUT&Tag-seq data were deposited into NCBI Gene Expression Omnibus database (GEO)[112] under the accession number GSE199419, GSE199366, and GSE214535, respectively. The proteomic data (MS raw data and peptide information) were deposited to the ProteomeXchange Consortium via the PRIDE[113] partner repository with the dataset identifier PXD032834 and 10.6019/PXD032834. The analysis package for RNA splicing, including all custom-designed scripts, was deposited in Zenodo under the accession number 7979503. The newly generated plasmids (pHD22Y/PfPRMT5-PTP and pHD22Y/ΔPfPRMT5) in this paper were deposited in

Addgene with ID numbers 203160 and 203161, respectively. The source data behind the graphs in the paper are included in Supplementary Data 1-8, including the processed microarray data (Supplementary Data 1), the processed CUT&Tag-seq data for H3R2me2s and PfPRMT5 (Supplementary Data 2 and 3), PfPRMT5 associated proteins (Supplementary Data 4), the alternative splicing events in ΔPfPRMT5 (Supplementary Data 5), the Known PRMT5 substrates from the published data (Supplementary Data 6), primers used in this study (Supplementary Data 7), and the source data behind Figs. 2 and 5, Supplementary Fig. 3, 4, 8c, and 10i (Supplementary Data 8). The uncropped and unedited blot/gel images corresponding to the blot/gel images were included as Supplementary Fig. 12 in the Supplementary information pdf file.

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

## Acknowledgements

We want to thank Richard Jones at MS Bioworks and Dale Chaput at the USF Proteomics Core facility for their assistance with the proteomic analysis. We thank the University of South Florida Genomics Program (Sequencing Core and Computational Core/Omics Hub) for supporting NGS sequence and data analysis. L. C. was supported by grants R01AI128940 and U19AI089672 from the National Institute of Allergy and Infectious Diseases, NIH, USA. J.M. was also supported by the startup fund from Morsani College of Medicine, University of South Florida, and grant R21AI149202 from the National Institute of Allergy and Infectious Diseases, NIH, USA.

## Author contributions

J.M. and L.C. conceived and designed the study. J.M., AB.L., and M.L. performed research, and acquired and analyzed data. C.W., Z.C., R.L., SR.A. and R.J. participated in the analysis of microarray, amplicon-seq and RNA-seq data. C.W. performed the alternative splicing analysis. X.Liang, H.M., Q. F., FA.S. assisted research. X.Li conducted parasite culture and phenotypic growth analysis. J.M. wrote the original draft. L.C. and X.C. revised the manuscript.

## Competing interests

The authors declares no competing interests.
