## [Peer Review File · Communications Biology]

Reviewers' comments:

Reviewer #1 (Remarks to the Author):

Lucky et al aims to investigate the function of type II PRMTs, which is in general known to introduce symmetric dimethylation at arginine residues in the histone and non-histone proteins across many species. The author found similar activity in the human malaria parasite *P. falciparum* as well. The conditional deletion of PfPRMT5 leads to defects in parasite invasion into RBCs. Further, the authors have reported that PfPRMT5 is associated with invasion-related transcriptional regulators such as AP2, GCN5, and BDP (It is not clear with the interaction, please refer to the specific comments). The PRMT5, known to methylate at H3R2 and H4R3 positions, which are conserved across many organisms, and it is not surprising or it is not a new finding that the PfPRMT5 introduces the gene activation marks in the malaria parasite. Although the data are quite interesting, the key experiments to support the PRMT5-mediated regulation of invasion-related genes are missing, which include ChIP for PfPRMT5, which will confirm that PRMT5-mediated regulation of invasion-related genes in the malaria parasite.

Major points:

(1) Figure 1: The authors have used human mononucleosomes as a substrate to test the methylation activity of endogenous PRMT5. They did not find methylation activity with histone substrates (No data was provided) and found PfPRMT methylates only synthetic mononucleosomes. The author should explain why the PfPRMT5 shows differential substrate preference.

(2) The authors should note that there is a contradiction in their results. They have observed methylation activity by PfPRMT5 on H3R8me2s and H4R3 residues (fig 1E). However, there are no changes in the methylation levels of the H3R8me2s and H4R3 marks in Δ PRMT5 *P. falciparum* (Fig 2E).

(3) This suggests that PfPRMT is not the sole enzyme for these residues in the parasite. It is essential to provide ChIP seq data for PfPRMT5 to draw conclusions on the mechanisms of PRMT5-mediated regulation of invasion-related genes. This is key data that will support the PRMT5-mediated regulation of invasion genes.

(4) In the abstract, it was mentioned that PfPRMT5 interacts with the transcriptional regulator of BDB1, AP2, and GCN5 proteins, however, I did not find any data to support this conclusion. Fig 5, depicts the list of cytosolic and nuclear proteins enriched and further annotated based on their functions. But no data to support that PfPRMT5 that interacts with these proteins. The statement may misguide the readers. I agree that enrichment of H3R3me2s ChIP seq signal to AP1 genes, but this does not rule out the possibility of the co-existence of gene activation marks.

Minor points:

1. The Western blot images should be labeled with a marker.
2. All the IF images should be provided with a scale bar.
3. The authors state that PfPRMT5 deposits H3R8me2s and H4Rme2s marks in the discussion (line # 397). It is advisable that the author should remove the statement as the data is not supported well.

Reviewer #2 (Remarks to the Author):

The manuscript submitted by Lucky and colleagues describes multiple possible roles for the arginine methyltransferase PfPRMT5 in the human malaria parasite *Plasmodium falciparum*. Arginine methylation has recently been shown to be an important regulatory modification in several organisms, and the methylation of specific positions in histones H3 and H4 have been shown to influence gene

expression. However, these modifications have not been studied in malaria parasites. Transcriptional regulation in malaria parasites is thought to be more dependent on epigenetic modifications than is observed in model organisms, thus the characterization of a role for arginine methylation is potentially important. The specific methyltransferase described in this paper (PRMT5) has been shown to be important for H3R2, H3R8 and H4R3 methylation as well as a role in mRNA splicing in model organisms, and a function in mRNA splicing was previously proposed in *P. falciparum*. Here the authors use several assays, including a knockout of the PfPRMT5 gene, to confirm these functions in malaria parasites. Further, they show that disruption of PfPRMT5 results in an important reduction in expression of genes linked to merozoite invasion.

While the results are somewhat predictable, this manuscript provides solid data demonstrating an important role for this methyltransferase in methylating histone H3 as well as mRNA splicing in *P. falciparum*. In addition, the link between H3R2 methylation and the expression of invasion genes is novel and provides insights into how parasites control expression of this important subset of genes.

Some comments for the authors to consider:

1. The authors use IFA and Western blots to show when PfPRMT5 is expressed and where it is localized throughout the parasites replicative cycle. In Figure 1C, the authors can easily detect the protein in rings while in the Western blot in Figure 1A it is barely detectable. A control in Figure 1C of wildtype parasites in which the protein has not been tagged would alleviate any possible concerns about background/nonspecific signal from the antibodies.

2. It is interesting that the disruption of the PfPRMT5 gene resulted in a reduction in H3R2me2s but not a complete loss of this mark. The authors speculate that H3R2me2a is deposited by a different methyltransferase. Do they similarly think a different methyltransferase can deposit the H3R2me2s mark and is responsible for the residual signal in the knockout line? In Figure S4 the authors attempt to show the specificity of the antibodies to H3R2me2s, but only show that they don't recognize unmodified H3. Are there any data on the specificity of these various antibodies from the supplier to show that they do not cross-react, for example with H3R2me2a?

3. The authors mapped the genomic positions of the H3R2me2s mark by CUTandTag-seq, showing quite specific deposition of the mark at positions of transcriptional activation. It would be interesting to perform this analysis on a PfPRMT5-knockout line. If the residual H3R2me2s that remains after PfPRMT5 disruption is deposited by a different methyltransferase, one might expect a different localization across the genome, or a specific loss of the mark at sites of transcriptional activation.

4. The figures/legends could be improved. For example, Figure 1, panel E: I presume these are human mononucleosomes, however it is not stated in the legend. Figure 5 currently has two "C" panels and no panel labeled B. In Figure S8, there are two tracks shown for each chromosome (one in red and one in blue), but the legend does not describe what each represents (presumably the blue displays IgG control?).

Responses to the Reviewers' Comments

We highly appreciate the reviewers' careful, thoughtful, and constructive reviews, which greatly helped us improve the manuscript. Based on the reviewers' comments and suggestions, we have performed several experiments, modified and/or added some figures, and revised the text in the manuscript. Below we summarized all criticisms and provided our responses, most of which were reflected in the revised manuscript. The changes in the revised manuscript are underlined for the comfort of reading.

Reviewers' comments:

Reviewer #1 (Remarks to the Author):

Lucky et al aims to investigate the function of type II PRMTs, which is in general known to introduce symmetric dimethylation at arginine residues in the histone and non-histone proteins across many species. The author found similar activity in the human malaria parasite *P. falciparum* as well. The conditional deletion of PfPRMT5 leads to defects in parasite invasion into RBCs. Further, the authors have reported that PfPRMT5 is associated with invasion-related transcriptional regulators such as AP2, GCN5, and BDP (It is not clear with the interaction, please refer to the specific comments). The PRMT5, known to methylate at H3R2 and H4R3 positions, which are conserved across many organisms, and it is not surprising or it is not a new finding that the PfPRMT5 introduces the gene activations marks in the malaria parasite. Although the data are quite interesting, the key experiments to support the PRMT5-mediated regulation of invasions-related genes are missing, which include ChIP for PfPRMT5, which will confirm that PRMT5-mediated regulation of invasion-related genes in the malaria parasite.

Major points:

Question 1 (Q1): Figure 1: The authors have used human mononucleosomes as a substrate to test the methylation activity of endogenous PRMT5. They did not find methylation activity with histone substrates (No data was provided) and found PfPRMT methylates only synthetic mononucleosomes. The author should explain why the PfPRMT5 shows differential substrate preference.

Answer 1 (A1): We briefly discussed this in the discussion section (lines 445-447). Previous publications showed that PRMT5 needs its partners (such as MEP50, pICln, kinase RioK1, and Grg4) for its activity and substrate specificity (PMID: 28061334, PMID: 26612103, PMID: 32025719, PMID: 22169276). A study showed that human PRMT5/MEP50 methylated free H3, but only methylated H4 in the H3/H4 tetramers. It primarily methylated H3 in the mono- or oligo-nucleosome from HeLa cells, in contrast, it failed to methylate H3 in the recombinant nucleosome (PMID: 32025719). Human PRMT5/Grg4 methylated both H3 and H4 in the mononucleosome, in contrast, it only methylated H3 in the core histones (PMID: 22169276). We used the eluate of the PfPRMT5 IPs from parasite nuclei for *in vitro*

methylation assays. Besides primary PfPRMT5, this eluate also contains other proteins, which are potentially necessary for methylating histones in the context of nucleosomes, not free histones.

In addition to providing more explanation on this issue in the discussion section (lines 447-450), we included images to show that PfPRMT5 did not methylate bovine and *P.f* core histones (Fig. 1D) and added a brief explanation in the results section (lines 159-162).

Q2: The authors should note that there is a contradiction in their results. They have observed methylation activity by PfPRMT5 on H3R8me2s and H4R3 residues (fig 1E). However, there are no changes in the methylation levels of the H3R8me2s and H4R3 marks in Δ PRMT5 *P. falciparum* (Fig 2E).

A2: PfPRMT5 showed methylation activities on H3R2, H3R8, and H4R3 in the *in vitro* methylation assays. These histone marks are expected to be an active mark (H3R2me2s) and silence marks (H3R8me2s and H4R3me2s) based on previous studies in other eukaryotic cells. Interestingly, disruption of *PfPRMT5* only resulted in a significant reduction in the level of active mark-H3R2me2s, consistent with the findings that disruption of *PfPRMT5* led to transcriptional downregulation of hundreds of genes without upregulation of any genes.

We do not know the reason why H3R8me2s and H4R3me2s did not change after the disruption of *PfPRMT5*. We speculate that the following are possible reasons.

1. Silence marks (H3R8me2s and H4R3me2s) and the active mark (H3R2me2s) are deposited by two different PRMTs.
 - a). Fig.1F showed that silence marks (H3R8me2s and H4R3me2s) consistently appeared in all asexual stages but H3R2me3 primary appeared at a high level in the schizont stage, consistent with the expression of PfPRMT5 (Fig. 1A), indicating that these marks are regulated differently.
 - b). Disruption of *PfPRMT5* only caused the reduction of the active mark (H3R2me2s), suggesting that only PfPRMT5 modifies H3R2 while another unknown PRMT modifies the silence marks (H3R8me2s and H4R3me2s). Similarly, PRMT5 purified from murine embryonic stem (ES) cells could methylate histone H2A and H4 *in vitro*. However, there was no significant loss of H4R3me2s modification but a selective loss of H2AR3me2s modification after the depletion of PRMT5 in murine ES cells (PMID: 21159818). Another report showed that human H2AR3me2s did not change after the depletion of PRMT5 in human ES cells (PMID: 24477620). Human/murine H2A and H4 contain the same R3 motif but the malaria parasite H2A does not contain the R3 motif.
 - c). PfPRMT5 interactome study found that PfPRMT5 interacts with GCN5, ADA2, and several WD-domain-containing proteins, indicating PfPRMT5 is likely involved in gene activation, in agreement with the function of H3R2me2s in gene activation and the findings that H3R2me2s preferably bind a WD domain protein (WDR5) for gene activation (PMID: 22231400).
 - d). In agreement with the above finding, we recently checked the chromatin landscape of H3R2me2s by CUT&Tag-seq in the *PfPRMT5* disruptant (Δ *PfPRMT5*) and found that H3R2me2s signals were completely depleted in the chromatin (lines 306-312, Fig. 4A right panel).
 - e). In many cases, there are discrepancies between *in vitro* and *in vivo* studies. *In vitro* methylation activities of PfPRMT5 might not reflect *in vivo* function of PfPRMT5.

2. If PfPRMT5 can methylate H3R8 and H4R2 *in vivo*, there might be other PRMTs that can make up for the loss of H3R8me2s and H4R3me2s by disruption of PfPRMT5.

Based on the above data and thoughts, we revised the manuscript in the results (lines 230-231, 306-312) and discussion (486-508).

Q3: This suggests that PfPRMT is not the sole enzyme for these residues in the parasite. It is essential to provide ChIP seq data for PfPRMT5 to draw conclusions on the mechanisms of PRMT5-mediated regulation of invasion-related genes. This is key data that will support the PRMT5-mediated regulation of invasion genes.

A3: We agree with the reviewer and performed PfPRMT5 chromatin landscape analysis by CUT&Tag-seq (lines 315-323, Table S3, Fig. S8B). As we described in the manuscript, CUT&Tag-seq is a ChIP-orthogonal method with a higher sensitivity and lower background compared to ChIP-seq. After mapping and peak calling, we identified 334 PfPRMT5 peaks. 78% (261) of them were localized in the 5'UTRs of 28 genes (Table S3). These 28 genes consist of 23 *var* genes, calcium-dependent protein kinase 1 (*PfCDPK1*), dipeptidyl aminopeptidase 3 (*DPAP3*), DNA helicase PSH3, heat shock protein 70 (*HSP70x*), and acyl-CoA binding protein, isoform 2 (*ACBP2*) (Table S3). Intriguingly, *PfCDPK1* is the only gene that has a known function in merozoite invasion, and this gene was also found significantly downregulated after PfPRMT5 disruption in our transcriptomic study.

Compared to 1147 H3R2me2s peaks at 5' UTR regions of 783 genes (also identified by CUT&Tag-seq) (Table S2), 27 of 28 genes including all 23 *var* genes and *PfCDPK1* with PfPRMT5 peaks at their 5'UTRs are overlapped with the genes containing H3R2me2s peaks at their 5'UTRs (Table S3). The signals of H3R2me2s peaks in these shared genes are significantly higher than the rest of the H3R2me2s peaks (Fig. S8B). These results indicate that CUT&Tag-seq of PfPRMT5 could only identify the locations with high PfPRMT5 occupancy, which potentially creates higher H3R2me2s signals in the chromatin.

The above results are consistent with the published studies. It is well-known that histone modifiers are notoriously difficult to be studied by ChIP-seq because of their highly dynamic nature and relatively weak binding to the chromatin, such as GCN5 (PMID: 28918903). Previous studies on PRMT5 and its histone marks in eukaryotic cells were normally conducted by ChIP-seq of histone marks along with ChIP-PCR of PRMT5, suggesting that ChIP-seq of PRMT5 was difficult in those studies (PMID: 30885941, PMID: 24097435). Recent reports showed two separate ChIP-seq analyses that identified 2927 and 3049 genes whose promoters or gene bodies were enriched with PRMT5 reads in pre-B leukemia cell line and HeLa cell line, respectively (PMID: 30635341, PMID: 33953349). While a recent study showed ChIP-seq in chicken identified 16041 genes with H3R2me2s peaks in promoters or gene bodies (PMID: 32199949) and another ChIP-seq study showed that H3R2me2s peaks near transcriptional start sites (TSSs) were positively correlated with gene expression in pro-B cells, indicating that H3R2me2s regulates almost all activated genes (PMID: 22720264). Taken together, there are intrinsic difficulties for PRMT5 ChIP-ing than its histone marks.

To confirm that PfPRMT5 and H3R2me2s coordinately mediate the regulation of invasion genes, we performed ChIP-qPCR to check the enrichment of PfPRMT5 and H3R2me2s in the 5' UTRs of the three invasion genes (*AMA1*, *EBA175*, and *MSP1*) at schizont stage (active stage)

compared to ring stage (silence stage) (lines 339-355, Fig. 5). The transcription of these genes was reduced (significantly for *AMA1* and *EBA175*) after *PfPRMT5* disruption (Fig. 3E, Table S1). Compared to the signals of *PfPRMT5* and H3R2me2s in the ring stage, both signals are substantially enriched in the 5' UTRs of the selected genes at the schizont stage whereas a control gene (*VAR2CSA*) did not show enrichment in both stages (Fig. 5 and S8C). The enrichment of H3R2me2s signals between schizont and ring stages were in most cases higher than that of *PfPRMT5*, again suggesting that ChIP-ing for *PfPRMT5* is more difficult than H3R2me2s.

These ChIP-qPCR results are somehow in agreement with the H3R2me2s CUT&Tag-seq data. H3R2me2s signals in the 5'UTRs of *MSP1*, *AMA1*, and *EBA175* were detected in all three (*MSP1*) or two replicates (*AMA1* and *EBA175*) of CUT&Tag-seq, respectively (Fig. S9). However, only one peak for each gene was called by SEACR program likely because the read counts in some of the replicates were not high enough to confidently be called peaks. Because our criteria used for defining real peaks are the presence in at least two of three replicates with at least 50% overlap of the peaks, these single peaks were excluded from the identified peaks.

Besides CUT&Tag-seq for *PfPRMT5*, we also performed CUT&Tag-seq for H3R2me2s in the *PfPRMT5* disruptant (*ΔPfPRMT5*) and the results showed that H3R2me2s signals were completely depleted in the chromatin (lines 306-312, Fig. 4A right panel).

Higher signals of *PRMT5* and H3R2me2s in the 5'UTRs of certain *var* genes suggest their potential function in regulating these genes in the subtelomeric regions. Although we noticed that *var* genes were significantly changed from our microarray, RNA-seq, and CUT&Tag-seq data, these data cannot be used for further in-depth analysis because our parasites are mixed clones expressing different *var* genes due to long-term culture. Further cloning is needed to study the function of *PfPRMT5* and H3R2me2s in the regulation of these important genes.

Based on the above new data, we revised the manuscript in the results (lines 314-355), discussion (lines 476-486), material and methods (lines 773-775, 802-830), and added Fig. 5, S8B, S8C, S9, Table S3 and revised Fig. 4B.

Q4: a) In the abstract, it was mentioned that *PfPRMT5* interacts with the transcriptional regulator of BDB1, AP2, and GCN5 proteins, however, I did not find any data to support this conclusion. **b)** Fig 5, depicts the list of cytosolic and nuclear proteins enriched and further annotated based on their functions. But no data to support that *PfPRMT5* that interacts with these proteins. The statement may misguide the readers. **c)** I agree that enrichment of H3R3me2s ChIP seq signal to AP1 genes, but this does not rule out the possibility of the co-existence of gene activation marks.

A4: a) The evidence that *PfPRMT5* interacts with BDP1, AP2, and GCN5 was derived from IPs using the *PfPRMT5::PTP* parasites. BDP1, AP2, and GCN5 were in the list of *PfPRMT5*-associated proteins (Table S4) after the Significance Analysis of INTeractome using stringent criteria (a probability of > 95% and false discovery rate [FDR] of < 1%). We revised Table S4 to let readers find these proteins easily.

b) We agree with the reviewer that no data supports that *PfPRMT5* directly interacts with the proteins because in this study we identified *PfPRMT5*-associated proteins by affinity

purification. We have revised our statement to avoid any suggestion of direct binding between PfPRMT5 and these proteins.

c) the localization of H3R2me2s in the 5' UTR region of *AP2-I* is only one dimension of gene activation. We have shown that other active marks such as H3K9ac and H3K4me3 were co-localized at active genes (Fig. 4C-E).

Minor points:

1. The Western blot images should be labeled with a marker.

We added the protein markers accordingly.

2. All the IF images should be provided with a scale bar.

We added the scale bars accordingly.

3. The authors state that PfPRMT5 deposits H3R8me2s and H4Rme2s marks in the discussion (line # 397). It is advisable that the author should remove the statement as the data is not supported well.

We revised that statement (now at lines 462-465) by adding “*in vitro*” in the sentence “although we found that PfPRMT5 could deposit the putative repressive (H3R8me2s and H4R3me2s) and the active (H3R2me2s) histone marks *in vitro*, we only found evidence of a significant decrease of the H3R2me2s mark after *PfPRMT5* disruption.” and discussed the possible reason that only H3R2me2s was significantly reduced after *PfPRMT5* disruption (lines 486-508).

Reviewer #2 (Remarks to the Author):

The manuscript submitted by Lucky and colleagues describes multiple possible roles for the arginine methyltransferase PfPRMT5 in the human malaria parasite *Plasmodium falciparum*. Arginine methylation has recently been shown to be an important regulatory modification in several organisms, and the methylation of specific positions in histones H3 and H4 have been shown to influence gene expression. However, these modifications have not been studied in malaria parasites. Transcriptional regulation in malaria parasites is thought to be more dependent on epigenetic modifications than is observed in model organisms, thus the characterization of a role for arginine methylation is potentially important. The specific methyltransferase described in this paper (PRMT5) has been shown to be important for H3R2, H3R8 and H4R3 methylation as well as a role in mRNA splicing in model organisms, and a function in mRNA splicing was previously proposed in *P. falciparum*. Here the authors use several assays, including a knockout of the PfPRMT5 gene, to confirm these functions in malaria parasites. Further, they show that disruption of PfPRMT5 results in an important reduction in expression of genes linked to merozoite invasion.

While the results are somewhat predictable, this manuscript provides solid data demonstrating an important role for this methyltransferase in methylating histone H3 as well as mRNA splicing in *P. falciparum*. In addition, the link between H3R2 methylation and

the expression of invasion genes is novel and provides insights into how parasites control expression of this important subset of genes.

Some comments for the authors to consider:

Q1. The authors use IFA and Western blots to show when PfPRMT5 is expressed and where it is localized throughout the parasites replicative cycle. In Figure 1C, the authors can easily detect the protein in rings while in the Western blot in Figure 1A it is barely detectable. A control in Figure 1C of wildtype parasites in which the protein has not been tagged would alleviate any possible concerns about background/nonspecific signal from the antibodies.

A1. Thanks a lot for the reviewer's pointing out this discrepancy. Indeed, IFA especially by using secondary antibodies will amplify signals, which could lead to a discrepancy. We re-performed IFA and took pictures of iRBCs at a suitable time of exposure to light from fluorescence microscopy and made sure the negative control (wild-type parasites) did not show any background/nonspecific signals from the antibodies (anti-protein C and Alexa fluor 488-conjugated IgG secondary antibodies). The new panel of IFA pictures agrees with the expression pattern shown by the Western blots.

Q2. 1) It is interesting that the disruption of the PfPRMT5 gene resulted in a reduction in H3R2me2s but not a complete loss of this mark. The authors speculate that H3R2me2a is deposited by a different methyltransferase. Do they similarly think a different methyltransferase can deposit the H3R2me2s mark and is responsible for the residual signal in the knockout line? **2)** In Figure S4 the authors attempt to show the specificity of the antibodies to H3R2me2s, but only show that they don't recognize unmodified H3. Are there any data on the specificity of these various antibodies from the supplier to show that they do not cross-react, for example with H3R2me2a?

A2. 1) Asymmetric (a) and symmetric (s) di-methylation of arginine are modified by Type I and Type II PRMT, respectively. In *P. falciparum*, only three PRMTs (PfPRMT1, PRMT4/CARM1, and PfPRMT5) were identified. PfPRMT1 is a Type I PRMT (PMID: 19344311) and PfPRMT5 is the only Type II PRMT based on the domain structure. Therefore, we did not speculate that the reduction in H3R2me2s but not a complete loss of this mark is because of another PRMT. We thought that the unspecific reaction of anti-H3R2me2s antibodies to unmethylated H3 is the reason that this mark was not completely lost especially when the blots were exposed to the substrate for a longer time. As shown in Fig. S4C, anti-H3R2me2s antibodies have a weak cross-reaction with unmethylated H3. This is a common issue for histone antibodies. To rule out the cross-reaction, we redid the WB by using recombinant histone without any modification as a negative control, lowering the loading of the parasite histones to make sure no cross-reaction band appeared in the negative control. By using this control, we obtained a new WB showing a very faint band in Δ PfPRMT5 parasites (Fig. 2E).

Based on the reviewer's suggestion (see **Q3** below), we also analyzed the chromatin landscape of H3R2me2s in Δ PfPRMT5 parasites by CUT&Tag-seq. We found that H3R2me2s signals were completely depleted in the chromatin. These results further validate that H3R2me2s is removed after PfPRMT5 disruption (see details in **A3**).

2) According to Reviewer's advice, we performed more Dot blots to assess whether there is any cross-reaction of anti-H3R2me2a to H3 tails with or without H3R2me2s modification. The results showed no cross-reactions to H3 peptides (Fig S4D).

Q3. The authors mapped the genomic positions of the H3R2me2s mark by CUTandTag-seq, showing quite specific deposition of the mark at positions of transcriptional activation. It would be interesting to perform this analysis on a PfPRMT5-knockout line. If the residual H3R2me2s that remains after PfPRMT5 disruption is deposited by a different methyltransferase, one might expect a different localization across the genome, or a specific loss of the mark at sites of transcriptional activation.

A3. This is excellent advice. We conducted 3 replicates of H3R2me2s CUT&Tag-seq in *PfPRMT5* disruptant ($\Delta PfPRMT5$) at the schizont stage. By comparing the signals from their IgG control, no or only low numbers (17 and 12) of peaks were called from three replicates. No real peaks of H3R2me2s were identified in the *PfPRMT5* disruptant ($\Delta PfPRMT5$) using the criteria of the presence in at least two of three replicates with at least 50% overlap of the peaks. H3R2me2s peaks in the WT parasites were substantially reduced to almost zero in *PfPRMT5* disruptant ($\Delta PfPRMT5$) (Fig. 4B right panel).

We revised the manuscript by adding these results (lines 306-312) in the manuscript.

Based on Reviewer #1's suggestions, we also performed CUT&Tag-seq for PfPRMT5 and ChIP-qPCR for checking the enrichment of PRMT5 and H3R2me2s in the 5' UTRs of three invasion genes (*MSP1*, *AMA1*, and *EBA175*). Please see these data in lines 314-355 and 476-485.

Q4. The figures/legends could be improved. For example, Figure 1, panel E: I presume these are human mononucleosomes, however it is not stated in the legend. Figure 5 currently has two "C" panels and no panel labeled B. In Figure S8, there are two tracks shown for each chromosome (one in red and one in blue), but the legend does not describe what each represents (presumably the blue displays IgG control?).

A4. Thanks for the reviewer's suggestion, we revised the figures/legends accordingly.

REVIEWERS' COMMENTS:

Reviewer #1 (Remarks to the Author):

The authors addressed all my questions. The revised manuscript was improved with additional data sets. I recommend the revised manuscript can be accepted for publication.

Reviewer #2 (Remarks to the Author):

The authors have addressed all the comments from my previous review. In particular, they added a description of additional chromatin localization studies to further validate their model. I have no additional recommendations.